# Constraint-Aware Flow Matching via Randomized Exploration

**Zhengyan Huan**                                           *Zhengyan.Huan@tufts.edu*
*Department of Electrical and Computer Engineering*
*Tufts University*
*NSF AI Institute for Artificial Intelligence and Fundamental Interactions*

**Jacob Boerma**                                               *Jacob.Boerma@tufts.edu*
*Department of Computer Science*
*Tufts University*

**Li-Ping Liu**                                                 *Liping.Liu@tufts.edu*
*Department of Computer Science*
*Tufts University*

**Shuchin Aeron**                                              *shuchin@ece.tufts.edu*
*Department of Electrical and Computer Engineering*
*Tufts University*
*NSF AI Institute for Artificial Intelligence and Fundamental Interactions*

**Reviewed on OpenReview:** *https://openreview.net/forum?id=OR4h9WPJhV*

## Abstract

We consider the problem of designing constraint-aware flow matching (FM) models that address the issue of constraint violations commonly observed in vanilla generative models. We consider two scenarios, viz.: (a) when a differentiable distance function to the constraint set is given, and (b) when the constraint set is only available via queries to a membership oracle. For case (a), we propose a simple adaptation of the FM objective with an additional term that penalizes the distance between the constraint set and the generated samples. For case (b), we propose to employ randomization and learn a mean flow that is numerically shown to have a high likelihood of satisfying the constraints. This approach deviates significantly from existing works that require simple convex constraints, knowledge of a barrier function, or a reflection mechanism to constrain the probability flow. Furthermore, in the proposed setting we show that a two-stage approach, where both stages approximate the same original flow but with only the second stage probing the constraints via randomization, is more computationally efficient than the corresponding one-stage approach. Through several synthetic cases of constrained generation, we numerically show that the proposed approaches achieve significant gains in terms of constraint satisfaction while matching the target distributions. As a showcase for a practical oracle-based constraint, we show how our approach can be used for training an adversarial example generator, using queries to a hard-label black-box classifier. We conclude with several future research directions. Our code is available at `https://github.com/ZhengyanHuan/FM-RE`.

## 1 Introduction

Implicit generative modeling, wherein one trains a neural network that transforms the input samples to samples potentially distributed according to the target distribution, has been at the forefront of modern AI and ML applications. Among the most successful are those rooted in the well-established theoretical framework of stochastic differential equations and probability flow ordinary differential equations (ODEs), namely Score-Based Models (Song et al., 2021), Flow Matching (Lipman et al., 2023; Albergo et al., 2023;

Liu et al., 2023b), Bridge Matching (Peluchetti, 2023), and Denoising Diffusion Probabilistic Models (Ho et al., 2020).

In many applications apart from matching the distribution of the data, it is also desired that the generated data does not violate constraints. In this context, there are two types of constraints that are considered, viz., (a) sample-wise constraints (Liu et al., 2023a; Feng et al., 2025; Lou & Ermon, 2023; Xie et al., 2024; Fishman et al., 2023; Christopher et al., 2024), and (b) distributional constraints (Khalafi et al., 2024). In this work, we consider the problem of learning to generate while satisfying sample-wise constraints. These sample-wise constraints are important and arise in a number of applications, such as watermark generation (Liu et al., 2023a), fluid dynamics (Feng et al., 2025), and image generation with certain attributes.

In this context, we build upon the versatile framework of FM and make the following **main contributions**.

1. We formalize FM with constraints under two cases, viz.: (a) specification via a differentiable distance to the constraint set, and (b) specification via a membership oracle that can be queried during training. Setting (b) departs from the specific cases considered thus far in the literature (**see section 5 for contrast with related works**), but it covers all of them. For (b) we propose a randomization strategy to enable learning of a mean flow for the constrained FM setup.

2. For numerical efficiency, we further derive a two-stage approach. Both stages aim to match the flow, while only the second stage is jointly optimized through a randomized exploration of the constraint set for constraint satisfaction.

3. On several synthetic examples, including non-convex, disconnected, and empty-interior constraints, our approach demonstrates the capability of both satisfying constraints and matching distributions.

4. We show several applications of the method for constrained image generation as well as generating adversarial examples for hard-label black-box image classifiers.

## 2 Problem Setup

Let $\mathcal{C} \subset \mathbb{R}^d$ denote a constraint set. We are given $n$ data points in $\mathbb{R}^d$ drawn i.i.d. from some unknown data distribution $q$, which is constrained on the set $\mathcal{C}$. That is, $\text{supp}(q) \subseteq \mathcal{C}$ where $\text{supp}(q)$ is the closure of the set of all points $x \in \mathbb{R}^d$ such that $q(x) > 0$. We consider the problem of learning to generate further samples from $q$. Therefore, the objective can be given as

$$\text{Generate } X \sim q, \quad \text{s.t. } X \in \mathcal{C}. \tag{1}$$

The $X \in \mathcal{C}$ requirement seems trivial since $\text{supp}(q) \subseteq \mathcal{C}$ is already assumed. However, conventional generative models often create samples that fail to obey this rule because $q$ is only observed via the samples, $\text{supp}(q)$ may not cover the entire constraint set, and $\mathcal{C}$ is often implicitly satisfied, which in high-dimensional settings makes the problem highly non-trivial. In this paper, for a given sample $x \in \mathbb{R}^d$, we consider two cases for constraint specification:

1. A differentiable distance between $x$ and $\mathcal{C}$ is known and can be given as

$$d(x, \mathcal{C}) = \inf_{z \in \mathcal{C}} \|x - z\|, \tag{2}$$

for some norm $\| \cdot \|$. For simplicity we will take this to be the Euclidean norm.

2. Only access to a query oracle that outputs $\mathbf{1}_{\mathcal{C}}(x)$ is given. Here $\mathbf{1}_{\mathcal{C}}(x) = 1$ if $x \in \mathcal{C}$, and 0 otherwise, denotes the usual indicator function of the constraints $\mathcal{C}$.

We note that $d(\cdot, \mathcal{C})$ is available in cases when $\mathcal{C}$ is convex or is a smooth manifold, e.g., a subspace, or other simple cases such as the ones considered in Liu et al. (2023a); Xie et al. (2024). The membership oracle is available or can be efficiently implemented for almost all constraints.

**Background on flow matching:** Define a stochastic process $X_t = \Psi_t(X_0, X_1) \in \mathbb{R}^d$ on $t \in [0,1]$ where the pair $(X_0, X_1) \sim \pi$ with marginals $q_0, q_1$ and where $\Psi_t(x_0, x_1) : [0,1] \times \mathbb{R}^d \times \mathbb{R}^d \rightarrow \mathbb{R}^d$ defines paths in $\mathbb{R}^d$ that are twice differentiable in space and time and uniformly Lipschitz in time satisfying $\Psi_0(x_0, x_1) = x_0, \Psi_1(x_0, x_1) = x_1$. The stochastic process $X_t = \Psi_t(X_0, X_1)$ is referred to as a *stochastic interpolant* between $q_0, q_1$ in Albergo et al. (2023). Notably, $\Psi_t(x_0, x_1)$ can be selected by the *user*. In this paper, we will only consider deterministic paths but one may also consider stochastic paths connecting the end points $x_0, x_1$. See Albergo et al. (2023) for examples of such constructions.

Following Albergo et al. (2023); Liu et al. (2023b); Albergo & Vanden-Eijnden (2023), define the *rectified* velocity field via $v^\Psi(z, t) = \mathbb{E}[\frac{d}{dt} X_t | X_t = z] = \mathbb{E}[\frac{d}{dt} \Psi_t | X_t = z]$. Then it can be shown that under some technical conditions (Albergo & Vanden-Eijnden, 2023), the stochastic process $Z_t$ that is driven by $v^\Psi(z, t)$ via the ODE $\frac{d}{dt} Z_t = v^\Psi(Z_t, t)$ that the time marginals of $Z_t$ and $X_t$ are equal in distribution (see Albergo & Vanden-Eijnden (2023) for a proof).

Therefore, for generative modeling purposes, one learns to match the rectified flow via $\arg\min_\theta \int_0^1 \mathbb{E}[\|u_\theta(X_t, t) - v^\Psi(X_t, t)\|^2] dt$, where the minimization is carried out over parameterized velocity fields $u_\theta : \mathbb{R}^d \times [0,1] \rightarrow \mathbb{R}^d$ parametrized by parameters $\theta$. As such this is an intractable optimization objective since it would entail simulating entire trajectories and approximating the conditional expectation of the true velocity field. But one can arrive at the following equivalence (Albergo et al., 2023; Liu et al., 2023b; Albergo & Vanden-Eijnden, 2023):

$$\arg\min_\theta \int_0^1 \mathbb{E}[\|u_\theta(X_t, t) - v^\Psi(X_t, t)\|^2] dt = \arg\min_\theta \int_0^1 \mathbb{E}[\|u_\theta(X_t, t) - \frac{d}{dt} \Psi_t(X_0, X_1)\|^2] dt,$$

where we recall that in the equation above $X_t = \Psi_t(X_0, X_1)$. Indeed given $\Psi_t$ and samples from $q_0, q_1$ one can efficiently approximate the latter objective via Monte-Carlo. This forms the basis of the general idea behind FM started by the seminal work Lipman et al. (2023) that was improved with optimal transport based couplings in Tong et al. (2024). The learned $u_\theta$ can then be used to generate samples from $q_1$ starting with samples from $q_0$.

## 3 Constraint-Aware Flow Matching

In this paper, while we can pick among many choices of $\Psi_t(x_0, x_1)$ and the couplings with marginals $q_0, q_1$, we choose to work with the simplest: linear interpolants $\Psi_t(x_0, x_1) = tx_1 + (1-t)x_0$ and the product coupling $q_0 \otimes q_1$ implying that $X_0 \perp X_1$.

Let $x_t^\theta$ be the (unique) solution to the ODE $\frac{d}{dt} z(t) = u_\theta(z(t), t), z(0) = x_0$. We define the general constraint-aware flow matching (CAFM) problem as solving for:

$$\arg\min_\theta \left\{ \int_0^1 \mathbb{E}[\|u_\theta(X_t, t) - \frac{d}{dt} \Psi_t(X_0, X_1)\|^2] dt - \lambda \mathbb{E}[\mathbf{1}_\mathcal{C}(X_1^\theta)] \right\}, \tag{3}$$

where $\mathcal{C}$ is the constraint set, $\lambda > 0$, and $\mathbf{1}_\mathcal{C}(\cdot)$ denotes the indicator function. We note that if one matches the flow exactly, then by assumption that $\text{supp}(q_1) \subseteq \mathcal{C}$, $\mathbb{E}[\mathbf{1}_\mathcal{C}(X_1^\theta)] = \mathbb{P}(X_1^\theta \in \mathcal{C}) = 1$. But in practice, due to the finite capacity of the parameters to capture velocity fields and due to approximation via limited samples, the learned $u_\theta$ will not exactly match the original flow and hence one needs to drive it to also satisfy the constraints.

In cases where a distance function $d(\cdot, \mathcal{C})$ available, an alternative formulation for CAFM is:

$$\arg\min_\theta \left\{ \int_0^1 \mathbb{E}[\|u_\theta(X_t, t) - \frac{d}{dt} \Psi_t(X_0, X_1)\|^2] dt + \beta \, \mathbb{P}(d(X_1^\theta, \mathcal{C}) \geq \varepsilon) \right\}, \tag{4}$$

for some $\beta > 0$ and for some $\varepsilon > 0$. Note that by the Markov inequality that $\mathbb{P}(d(X_1^\theta, \mathcal{C}) \geq \varepsilon) \leq \frac{\mathbb{E}[d(X_1^\theta, \mathcal{C})]}{\varepsilon}$ and therefore we can choose to instead solve:

$$\arg\min_\theta \left\{ \int_0^1 \mathbb{E}[\|u_\theta(X_t, t) - \frac{d}{dt} \Psi_t(X_0, X_1)\|^2] dt + \lambda \, \mathbb{E}[d(X_1^\theta, \mathcal{C})] \right\}, \tag{5}$$

for some $\lambda = \beta/\varepsilon > 0$. Note that Markov's inequality allows for error, so the objective in equation 5 has a possibly different minimum to that in equation 4.

The two cases equation 3 and equation 5 are fundamentally different for training purposes. While in the first case the constraints can only be probed by the flow via a membership oracle, in the second case the distance function directly yields a proxy for the probability that the constraints are violated. In the next section 4 we present methods for both cases and for case equation 3, we propose to use randomization in the flow to explore the constraints, yielding a mean flow for constrained FM.

## 4 Methods

### 4.1 Constraint-Aware Flow Matching with Differentiable Distance (FM-DD)

Following the objective in equation 5, the training and sampling algorithms for FM-DD are given in Alg. 1 and Alg. 2. Note that we fix a forward Euler discretization:

$$X_{t+\Delta t}^{\theta} = X_t^{\theta} + u_\theta(X_t^{\theta}, t)\Delta t, \tag{6}$$

with $X_0^{\theta} \sim p_0$. $\Delta t \ll 1$ is a selected interval for discretization. The sample generated according to this procedure is $X_1^{\theta}$.

---

**Algorithm 1** FM-DD training

**Input:** $u_\theta$, $\Delta t$, $N$, $d(\cdot, \mathcal{C})$, $\lambda$, $q_1$, learning rate $\eta$, batch size $B$
**Output:** $u_\theta$

**repeat**
    $\mathcal{L} \leftarrow 0$
    **for** $b = 1, 2, \cdots, B$ **do**
        Obtain $x_1^\theta$ according to Alg. 2
        $x_0 \sim \mathcal{N}(\mathbf{0}, \boldsymbol{I})$, $x_1 \sim q_1$
        $i \sim \text{Uniform}([0 : N - 1])$, $t \leftarrow i\Delta t$
        $\psi_t(x_0, x_1) \leftarrow (1 - t)x_0 + tx_1$
        $\mathcal{L} \leftarrow \mathcal{L} + \|u_\theta(\psi_t(x_0, x_1), t) - (x_1 - x_0)\|^2 + \lambda d(x_1^\theta, \mathcal{C})$
    **end for**
    $\theta \leftarrow \theta - \eta\nabla_\theta\mathcal{L}$
**until** converged
**Return** $u_\theta$

---

**Algorithm 2** FM-DD sampling

**Input:** $u_\theta$, $\Delta t$
**Output:** $x_1$

$x_0 \sim \mathcal{N}(\mathbf{0}, \boldsymbol{I})$
$t \leftarrow 0$
**repeat**
    $x_{t+\Delta t} \leftarrow x_t + u_\theta(x_t, t)\Delta t$
    $t \leftarrow t + \Delta t$
**until** $t = 1$
**Return** $x_1$

---

### 4.2 Constraint-Aware Flow Matching via Randomized Exploration (FM-RE)

We now deal with the objective in equation 3 without an available differentiable distance, instead, only the membership oracle of constraint satisfaction is accessible. In a conventional FM model, the output $X_1^\theta$ is deterministic given the initial point $X_0$. As a result, the constraint term $\mathbb{E}[\mathbf{1}_{\mathcal{C}}(X_1^\theta)]$ reduces to an indicator function evaluated at a deterministic point. Since the gradient of an indicator function is 0 almost everywhere except on the boundary, the gradient with respect to $\theta$ vanishes almost everywhere. One way to address this issue is to introduce randomness into the sampling process, so that $X_1^\theta$ becomes a random variable rather than a deterministic output. The constraint term then corresponds to a probability of satisfaction, yielding a differentiable expectation with respect to $\theta$ and enabling gradient-based optimization. To this end, we propose a method we refer to as FM-RE, which learns a flow in two parts. First, a flow is learned via vanilla FM. Then, for time steps after a given $t_0$, randomization is used to explore the constraints available via the membership oracle and derive a mean flow that has a high likelihood to satisfy the constraints. We note that this approach is used frequently in stochastic control and reinforcement learning (Sutton & Barto, 2018).

Instead of a deterministic velocity, we pick a randomized velocity in equation 3, defined as a random variable $\tilde{U}^{\theta,\sigma}(X_t, t, W_\sigma(t))$, where $W_\sigma(t)$ is a random variable that depends only on time $t$, is independent of $X_t$, and is parameterized by parameters $\sigma$. Thus, the objective becomes:

$$\arg\min_{\theta,\sigma}\left\{\int_0^1 \mathbb{E}[\|\tilde{U}_{\theta,\sigma}(X_t,t,W_\sigma(t)) - \frac{d}{dt}\Psi_t(X_0,X_1)\|^2]dt - \lambda\mathbb{E}[\mathbf{1}_{\mathcal{C}}(X_1^{\theta,\sigma})]\right\},$$

$$\text{with}\quad \frac{d}{dt}X_t^{\theta,\sigma} = \tilde{U}_{\theta,\sigma}(X_t,t,W_\sigma(t)),\ X_0^{\theta,\sigma} \sim q_0. \tag{7}$$

As a quick observation, by appealing to Jensen's inequality, we note that:

$$\mathbb{E}\left[\left\|\tilde{U}_{\theta,\sigma}(X_t,t,W_\sigma(t)) - \frac{d}{dt}\Psi_t(X_0,X_1)\right\|^2\right] = \mathbb{E}_{X_t}\left[\mathbb{E}_{W_\sigma(t)}\left[\left\|\tilde{U}_{\theta,\sigma}(X_t,t,W_\sigma(t)) - \frac{d}{dt}\Psi_t(X_0,X_1)\right\|^2\right]\right]$$

$$\geq \mathbb{E}_{X_t}\left[\left\|\mathbb{E}_{W_\sigma(t)}\left[\tilde{U}_{\theta,\sigma}(X_t,t,W_\sigma(t))\right] - \frac{d}{dt}\Psi_t(X_0,X_1)\right\|^2\right]. \tag{8}$$

Therefore, if $\mathbb{E}_{W_\sigma(t)}\left[\tilde{U}_{\theta,\sigma}(X_t,t,W_\sigma(t))\right]$ is viewed as a mean flow, the FM objective part in equation 7 provides an upper bound for its FM loss. In this work, we specifically pick

$$\tilde{U}_t^{\theta,\sigma} = \tilde{U}_{\theta,\sigma}(X_t,t,W_\sigma(t)) = u_\theta(X_t,t) + \sigma_t W, \tag{9}$$

where $W \sim \mathcal{N}(\mathbf{0},\boldsymbol{I}), \sigma_t \in \mathbb{R}^d$, and $\sigma$ is a collection of $\sigma_t$. The probability density function (PDF) of $\tilde{u}_t^{\theta,\sigma} \in \mathbb{R}^d$ based on $x_t^{\theta,\sigma}$ can be denoted as

$$\pi_{\theta,\sigma}(\tilde{u}_t^{\theta,\sigma}|x_t^{\theta,\sigma}) = \mathcal{N}(\tilde{u}_t^{\theta,\sigma}|u_\theta(x_t^{\theta,\sigma},t),\sigma_t^2\boldsymbol{I}). \tag{10}$$

Similar to the previous subsection, we work with a forward Euler discretization of the stochastic velocity field for training purposes, i.e., $X_{t+\Delta t}^{\theta,\sigma} = X_t^{\theta,\sigma} + \tilde{U}_t^{\theta,\sigma}\Delta t$. Pick an integer $N$ and let $\Delta t = \frac{1}{N}$. Define a random trajectory $\mathcal{T}$ with $N$ steps as follows,

$$\mathcal{T} = (X_0^{\theta,\sigma}, \tilde{U}_0^{\theta,\sigma}, X_{\Delta t}^{\theta,\sigma}, \tilde{U}_{\Delta t}^{\theta,\sigma}, X_{2\Delta t}^{\theta,\sigma}, \tilde{U}_{2\Delta t}^{\theta,\sigma}, \cdots, X_1^{\theta,\sigma}),$$

in which $X_0^{\theta,\sigma} \sim q_0$. We note the following Proposition, which is proved in the Appendix A.

**Proposition 4.1.** *Let $X_1(\mathcal{T})$ denote the generated sample of trajectory $\mathcal{T}$, i.e., $X_1^{\theta,\sigma}$. Under assumptions of all $X_t^{\theta,\sigma}$ having strictly positive density on $\mathbb{R}^d$, $\mathcal{C}$ being a subset of $\mathbb{R}^d$ with positive Lebesgue measure, existence and uniqueness of the solutions driven by the respective ODEs, boundedness of $\nabla_{\theta,\sigma}\log\pi_{\theta,\sigma}(\tilde{U}_{i\Delta t}^{\theta,\sigma}|X_{i\Delta t}^{\theta,\sigma})$, and finiteness of all the expectations involved:*

$$\nabla_{\theta,\sigma}\mathbb{E}\left[\mathbf{1}_{\mathcal{C}}(X_1^{\theta,\sigma})\right] = \mathbb{E}_{\mathcal{T}}\left[\sum_{i=0}^{N-1}\left(\nabla_{\theta,\sigma}\log\pi_{\theta,\sigma}(\tilde{U}_{i\Delta t}^{\theta,\sigma}|X_{i\Delta t}^{\theta,\sigma})\right)\mathbf{1}_{\mathcal{C}}(X_1(\mathcal{T}))\right].$$

We can approximate the expectation in Proposition 4.1 by sampling from $q_0$, simulating the trajectories with the forward Euler discretization of the ODE according to $\theta,\sigma$, and utilizing the closed-form expression for $\log\pi_{\sigma,\theta}$.

There are two numerical issues in approximating the expectation above. First, $N$ is typically very large to control the error in forward Euler approximation $O(\Delta t)$ (Ascher & Petzold, 1998), which leads to an increased backpropagation cost. Second, with a large $N$ one also has to randomize over a larger number of random variables, thus requiring more samples for estimation and a longer convergence time overall.

One way to mitigate the second issue is to introduce randomness only for $t \geq t_0$, i.e., $\sigma_t = 0$ for $t < t_0$ and $\sigma_t > 0$ for $t \geq t_0$, in which $t_0 \in (0,1)$ is a hyperparameter. Assuming that sampling a trajectory from $t = 0$ to $t = 1$ requires $N$ steps and there are $N_1$ steps before $t_0$ and $N_2$ steps after $t_0$, then we have $N_1 + N_2 = N$. The realization of a stochastic trajectory starts at $t = t_0$ and becomes $\tau = (x_{t_0}^{\theta,\sigma}, \tilde{u}_{t_0}^{\theta,\sigma}, x_{t_0+\Delta t}^{\theta,\sigma}, \tilde{u}_{t_0+\Delta t}^{\theta,\sigma}, \cdots, x_1^{\theta,\sigma})$ with $N_2$ steps. The first $N_1$ steps are deterministic conditioned on $x_0^{\theta,\sigma} \sim q_0$, i.e., $x_{t_0}^\theta = x_{t_0}^{\theta,\sigma} = x_0^{\theta,\sigma} + \Delta t\sum_{i=0}^{N_1-1}u_\theta(x_{i\Delta t}^{\theta,\sigma},i\Delta t)$.

---

**Algorithm 3** FM-RE training

**Input:** $u_{\theta_1}$, $u_{\theta_2}$, $\sigma$, $t_0$, $\Delta t$, $N_2$, $\mathbf{1}_\mathcal{C}(\cdot)$, $q_1$, learning rate $\eta_1, \eta_2$, batch size $B_1, B_2$
**Output:** $u_{\theta_1}$, $u_{\theta_2}$, $\sigma$

**repeat**
    $\mathcal{L}_1 \leftarrow 0$
    **for** $b_1 = 1, 2, \cdots, B_1$ **do**
        $x_0 \sim \mathcal{N}(\mathbf{0}, \boldsymbol{I})$, $x_1 \sim q_1, t \sim \mathcal{U}[0,1]$
        $\psi_t(x_0, x_1) \leftarrow (1-t)x_0 + tx_1$
        $\mathcal{L}_1 \leftarrow \mathcal{L}_1 + \|u_{\theta_1}(\psi_t(x_0, x_1), t) - (x_1 - x_0)\|^2$
    **end for**
    $\theta_1 \leftarrow \theta_1 - \eta_1 \nabla_{\theta_1} \mathcal{L}_1$
**until** converged
$\theta_2 \leftarrow \theta_1$
**repeat**
    $\mathcal{L}_2 \leftarrow 0$
    **for** $b_2 = 1, 2, \cdots, B_2$ **do**
        Obtain $x_1(\tau) = x_1^{\theta_2, \sigma}$ via Alg. 4 (randomized)
        $\mathcal{L}_\mathcal{C} \leftarrow -\sum_{i=0}^{N_2-1} \left( \log \pi_{\theta_2, \sigma}(\tilde{u}_{t_0+i\Delta t}^{\theta_2,\sigma} | x_{t_0+i\Delta t}^{\theta_2,\sigma}) \right) \mathbf{1}_\mathcal{C}(x_1(\tau))$
        $x_0 \sim \mathcal{N}(\mathbf{0}, \boldsymbol{I})$, $x_1 \sim q_1$
        $i \sim \text{Uniform}([0 : N_2 - 1])$, $t \leftarrow t_0 + i\Delta t$
        $\psi_t(x_0, x_1) \leftarrow (1-t)x_0 + tx_1$
        $\tilde{u}_t^{\theta_2,\sigma} \sim \mathcal{N}(u_{\theta_2}(\psi_t(x_0, x_1), t), \sigma_t^2 \boldsymbol{I})$
        $\mathcal{L}_2 \leftarrow \mathcal{L}_2 + \|\tilde{u}_t^{\theta_2,\sigma} - (x_1 - x_0)\|^2 + \lambda \mathcal{L}_\mathcal{C}$
    **end for**
    $\{\theta_2, \sigma\} \leftarrow \{\theta_2, \sigma\} - \eta_2 \nabla_{\theta_2, \sigma} \mathcal{L}_2$
**until** converged
**Return** $u_{\theta_1}$, $u_{\theta_2}$, $\sigma$

---

**Algorithm 4** FM-RE sampling

**Input:** $u_{\theta_1}$, $u_{\theta_2}$, $\sigma$, $t_0$, $\Delta t$
**Output:** $x_1$

$x_0 \sim \mathcal{N}(\mathbf{0}, \boldsymbol{I})$, $t \leftarrow 0$
**repeat**
    $x_{t+\Delta t} \leftarrow x_t + u_{\theta_1}(x_t, t)\Delta t$
    $t \leftarrow t + \Delta t$
**until** $t = t_0$
**repeat**
    **if** randomized **then**
        $\tilde{u}_t \sim \mathcal{N}(u_{\theta_2}(x_t, t), \sigma_t^2 \boldsymbol{I})$
    **else**
        $\tilde{u}_t \leftarrow u_{\theta_2}(x_t, t)$
    **end if**
    $x_{t+\Delta t} \leftarrow x_t + \tilde{u}_t \Delta t$
    $t \leftarrow t + \Delta t$
**until** $t = 1$
**Return** $x_1$

---

**Proposition 4.2.** *For randomization starting at time $t_0 > 0$, under the assumptions of Proposition 4.1 we have:*

$$\nabla_{\theta,\sigma} \mathbb{E}\left[\mathbf{1}_\mathcal{C}(X_1^{\theta,\sigma})\right] = \mathbb{E}_\mathcal{T}[\nabla_{\theta,\sigma} \log p(X_{t_0}^\theta) \mathbf{1}_\mathcal{C}(X_1(\mathcal{T}))] + \mathbb{E}_\mathcal{T}\left[\sum_{i=0}^{N_2-1} \left(\nabla_{\theta,\sigma} \log \pi_{\theta,\sigma}(\tilde{U}_{t_0+i\Delta t}^{\theta,\sigma} | X_{t_0+i\Delta t}^{\theta,\sigma})\right) \mathbf{1}_\mathcal{C}(X_1(\mathcal{T}))\right].$$

The proof is provided in Appendix B. However, we are now faced with another computational challenge, namely, to approximate $\mathbb{E}_\mathcal{T}[\nabla_{\theta,\sigma} \log p(X_{t_0}^\theta) \mathbf{1}_\mathcal{C}(X_1(\mathcal{T}))]$, where it is evident that there is no access to an analytical form for $\log p(X_{t_0}^\theta)$. This can potentially be done by using non-parametric score estimators (Zhou et al., 2020) by generating a lot of samples or via computing the normalized probability flow corresponding to the compositions of the transformations corresponding to the forward Euler discretization (if it can be guaranteed that the transformations are invertible) (Papamakarios et al., 2021). Nevertheless, both approaches come with their own significant computational burden.

Instead, we adopt two sets of parameters $\theta_1$ and $(\theta_2, \sigma)$ for $t < t_0$ and $t \geq t_0$ respectively. The velocity $u_{\theta_1}(X, t)$ for $t < t_0$ is deterministic. The velocity $\tilde{U}_t^{\theta_2,\sigma}$ for $t \geq t_0$ becomes stochastic following equation 9. First $\theta_1$ is optimized with the FM objective via

$$\arg\min_{\theta_1} \int_0^{t_0} \mathbb{E}[\|u_{\theta_1}(X_t, t) - \frac{d}{dt}\Psi_t(X_0, X_1)\|^2]dt. \tag{11}$$

Note that $\nabla_{\theta_2,\sigma} \log p(X_{t_0}^{\theta_1}) = 0$. Then $\theta_1$ is frozen and $(\theta_2, \sigma)$ is optimized via

$$\arg\min_{\theta_2,\sigma} \left\{ \int_{t_0}^1 \mathbb{E}[\|\tilde{U}_t^{\theta_2,\sigma} - \frac{d}{dt}\Psi_t(X_0, X_1)\|^2]dt - \lambda \mathbb{E}\left[\mathbf{1}_\mathcal{C}(X_1^{\theta_2,\sigma})\right] \right\}, \tag{12}$$

in which $\nabla_{\theta_2,\sigma} \mathbb{E}\left[\mathbf{1}_\mathcal{C}(X_1^{\theta_2,\sigma})\right]$ is given as

$$\mathbb{E}_{\mathcal{T}}\left[\sum_{i=0}^{N_2-1}\left(\nabla_{\theta_2,\sigma}\log\pi_{\theta_2,\sigma}(\tilde{U}_{t_0+i\Delta t}^{\theta_2,\sigma}|X_{t_0+i\Delta t}^{\theta_2,\sigma})\right)\mathbf{1}_{\mathcal{C}}(X_1(\mathcal{T}))\right],$$

where $\mathcal{T}$ starts at $X_{t_0}^{\theta_2,\sigma}=X_{t_0}^{\theta_1}$ and has $N_2$ steps. The training algorithm is given in Alg. 3. Recall equation 8, we can regard $\mathbb{E}[\tilde{U}_t^{\theta_2,\sigma}]=u_{\theta_2}(X,t)$ as a deterministic mean velocity to remove randomness when sampling, as shown in Alg. 4. We emphasize that FM-RE's two-stage procedure is not a fine-tuning algorithm (Uehara et al., 2024), instead, the first stage trains for $t<t_0$ following FM objective, and the second stage for $t\geq t_0$ is a joint optimization framework, where both distributional similarity and constraint satisfaction are optimized under a unified objective. We further contrast this distinction and show that FM-RE achieves better trade-off in distribution matching vs. constraint satisfaction in Appendix C.1.1.

## 5 Related Works

| | DDPM, FM | RDM | RFM | MDM | NAMM | PDM | FM-DD | FM-RE |
|---|---|---|---|---|---|---|---|---|
| $\mathbb{P}(X_1\notin\mathcal{C})$ | $p_0$ | $0$ | $0$ | $0$ | $0<p\ll p_0$ | $0$ | $0<p\ll p_0$ | $0<p\ll p_0$ |
| Requirements on $\mathcal{C}$ | N/A | Convex, explicit knowledge of $\partial\mathcal{C}$ | Connected, explicit knowledge of $\partial\mathcal{C}$ | Convex, closed-form bijective projection | $d(\cdot,\mathcal{C})$, learnable bijective projection | Projection to $\mathcal{C}$ | $d(\cdot,\mathcal{C})$ | Membership (loosest) |

Table 1: A tabular summary of related works. $\mathbb{P}(X_1\notin\mathcal{C})$ represents the probability of constraint violations. Denoising diffusion probabilistic models (DDPM) and FM appear in Ho et al. (2020) and Lipman et al. (2023), respectively. Reflected flow matching (RFM) appears in Xie et al. (2024). Mirror diffusion model (MDM) appears in Liu et al. (2023a). Neural approximate mirror map (NAMM) appears in Feng et al. (2025). Projected diffusion model (PDM) appears in Christopher et al. (2024). **This work**: FM-DD and FM-RE.

We compare several related works with our methods in Table 1. The recent works that handle constrained generation are primarily reflection-based methods, such as the reflected diffusion model (RDM) (Lou & Ermon, 2023; Fishman et al., 2023) and reflected flow matching (RFM) (Xie et al., 2024) that provide solutions to enforce strict constraint satisfaction by designing paths such that $\Psi_t(x_0,x_1)\in\mathcal{C}\,\forall t\in[0,1]$. However, designing such paths requires knowledge of the normal to $\partial\mathcal{C}$. Mirror diffusion model (MDM) (Liu et al., 2023a) is another method to ensure that the generated samples satisfy the constraints. MDM constructs a bijective projection between the constraint set and an unconstrained domain. Learning is performed in the unconstrained domain, and samples are generated in the unconstrained domain before being mapped back to the constraint set via the inverse projection. However, such bijective projections in closed-form only exist for a subset of convex constraints. Neural approximate mirror map (NAMM) (Feng et al., 2025) aims to approximate the bijective projection in MDM via neural networks and expands the application domain to non-convex constraint sets. Nevertheless, this requires careful construction to ensure invertibility, especially when the training samples are limited and the constraint set is complex. Projected diffusion model (PDM) (Christopher et al., 2024) is an inference-time method for constraint-aware generation, i.e., it does not modify the training process of diffusion models, instead, it enforces constraints at inference time by projecting stochastic diffusion updates onto $\mathcal{C}$ at each step. This projection may alter the resulting sampled distribution. When a distance $d(\cdot,\mathcal{C})$ is available, one can utilize a simple adaptation of the FM, i.e., FM-DD, to penalize outliers. Although FM-RE is not able to ensure zero constraint violation as in reflection-based methods and MDM, it can achieve a much higher constraint satisfaction rate than basic generative models while still matching the distribution. Moreover, among the mentioned works, FM-RE is able to handle simple as well as complex constraints requiring only a constraint satisfaction oracle.

## 6 Experiments

We refer the reader to Appendix C for a detailed description of all the experimental setups and parameter configurations for reproducibility. Only the mean velocity, i.e., $\mathbb{E}[\tilde{U}_t^{\theta_2,\sigma}]=u_{\theta_2}(X,t)$ is employed in the sampling process of the second stage of FM-RE.

## 6.1 Synthetic Experiments

|  |  | Box | 2 boxes | 8d $\ell_2$ ball | 20d $\ell_2$ ball | Subspace |
|---|---|---|---|---|---|---|
| SWD ($\downarrow$) | FM | 0.1268 | 0.2260 | 0.0193 | 0.0087 | 0.0372 |
|  | RFM | 0.1258 | N/A | 0.0177 | 0.0098 | N/A |
|  | MDM | 0.1431 | N/A | 0.0292 | 0.0159 | N/A |
|  | NAMM | 0.1680 | 0.2101 | 0.0228 | 0.0205 | 0.2124 |
|  | PDM | 0.1268 | 0.5661 | 0.0200 | 0.0257 | 0.3079 |
|  | FM-DD | 0.1228 | 0.2174 | 0.0175 | 0.0086 | 0.0356 |
|  | FM-RE | 0.1250 | 0.2104 | 0.0194 | 0.0132 | 0.0355 |
| $\mathbb{P}(X_1 \notin \mathcal{C})$ (‰, $\downarrow$) | FM | 1.132 | 4.580 | 23.67 | 90.82 | 790.1 |
|  | RFM | 0 | N/A | 0 | 0 | N/A |
|  | MDM | 0 | N/A | 0 | 0 | N/A |
|  | NAMM | 0.387 | 0.379 | 0.138 | 2.890 | 707.6 |
|  | PDM | 0 | 0 | 0 | 0 | 0 |
|  | FM-DD | 0.053 | 0.073 | 0.140 | 0.502 | 86.24 |
|  | FM-RE | 0.066 | 0.222 | 0.768 | 2.513 | 98.58 |

Table 2: Performance comparison for synthetic experiments. Lower values indicate better performance for both metrics.

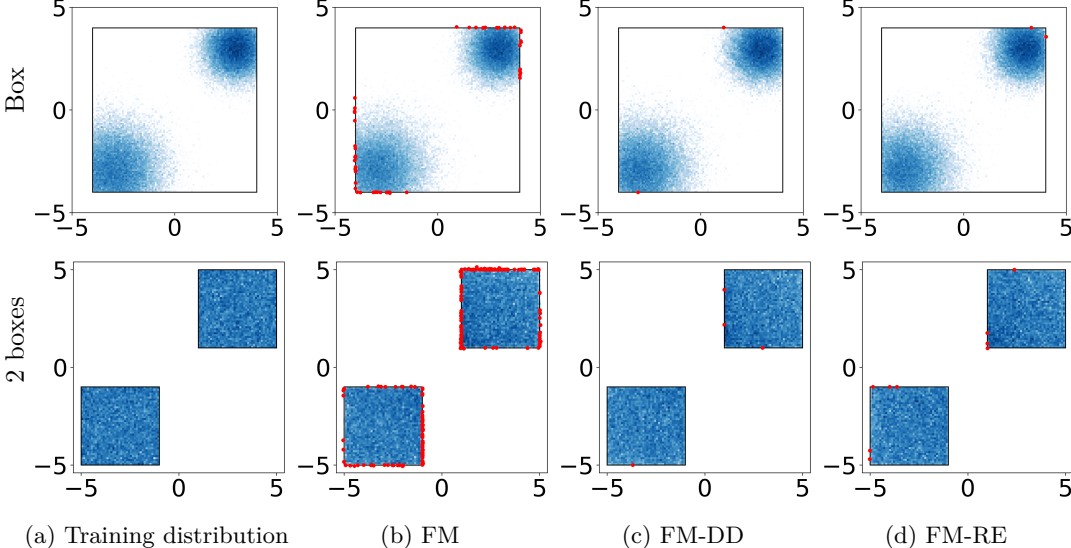

(a) Training distribution (b) FM (c) FM-DD (d) FM-RE

Figure 1: The histplots of samples generated by different methods compared to samples in the training distribution. Samples violating the constraints are highlighted in red. The total sample size is 50000.

For each synthetic case, the following approaches are compared: FM (Lipman et al., 2023), RFM (Xie et al., 2024), MDM (Liu et al., 2023a), NAMM (Feng et al., 2025), PDM (Christopher et al., 2024), FM-DD, and FM-RE. The details of the compared baselines are provided in Appendix C.1. We report the constraint violation rate and the Sliced Wasserstein Distance (SWD) (Bonneel et al., 2014; Rabin et al., 2012) averaged over 100 generation trials ($10^4$ samples each) in Table 2. SWD measures the similarity between two distributions, with smaller values indicating greater similarity. N/A means RFM and MDM cannot apply to some of the cases.

**2-D toy examples:** We first test the effectiveness and visualize the generation performance of the proposed methods on 2-D constraint sets, including a box constraint (**Box**) and a disconnected boxes constraint (**2 boxes**). Notably, the **2 boxes** constraint is non-convex and disconnected.

Fig. 1 shows the histplots of samples obtained from FM, FM-DD, and FM-RE. We can observe that all three methods can generate samples following the training distribution. However, FM-DD and FM-RE generate significantly fewer samples violating the constraints (denoted in red). The stats in Table 2 show that, for **Box**,

a convex constraint, all constraint-aware approaches achieve satisfactory performance. While for **2 boxes**, a disjoint constraint set, RFM and MDM are not applicable. PDM's sampled distribution deviates from the training distribution, leading to a large SWD metric. NAMM, FM-DD, and FM-RE generate samples following the training distribution, and satisfying the constraints with a higher probability compared with FM.

$\ell_2$ **ball constraints:** We next evaluate the proposed methods for higher-dimensional constraints, i.e., 8-$d$ and 20-$d$ $\ell_2$ ball constraints following Liu et al. (2023a). The target distribution is a Gaussian mixture. The results in Table 2 illustrate the superior constraint satisfaction performance of FM-DD and FM-RE over FM. In addition, for the $d = 20$ case, we vary $t_0 \in \{0, 0.2, 0.4, 0.6, 0.8\}$, $\lambda \in \{2, 5, 10, 20, 30\}$, $N_2 \in \{75, 60, 45, 30, 15\}$, and $N_1 = 75 - N_2$ to illustrate the relationship between distributional match and constraint satisfaction. The results are shown in Fig. 2, in which we can observe that FM-DD achieves the best performance using $d(\cdot, \mathcal{C})$. FM-RE achieves a much better constraint satisfaction rate than FM via queries and explorations. Increasing $\lambda$ can further reduce constraint violations at the cost of distributional match. FM-RE with different $t_0$ displays similar performances ; however, a larger $t_0$ requires a smaller $N_2$, leading to lighter computational cost. The training time required to complete the same number of iterations for $t_0 \in \{0, 0.2, 0.4, 0.6, 0.8\}$ is approximately $27 : 23 : 19 : 15 : 11$, respectively. These results provide numerical justification that setting $t_0 > 0$ can reduce training time and computational cost without substantially compromising performance.

**Subspace constraint:** Subspace constraint is a special type of constraint since it has no interior. Specifically, we consider a 10-D multivariate Gaussian distribution's projection to a 9-D hyperplane $\mathcal{C}$. The explicit mathematical expression of the hyperplane is unknown to the models, however, the distance of any sample to this hyperplane is available to them. Note that it is not likely for the generated samples to fall exactly in the 9-D hyperplane. For FM-RE, a sample is considered to satisfy the constraint if its distance to the hyperplane is smaller than a threshold $5 \times 10^{-4}$.

Table 2 shows that the FM model without constraint guidance usually generates samples with a further distance compared to the threshold. NAMM fails to learn an effective forward and backward function between the constraint set and the unconstrained space, leading to a deviation between the generated distribution and the training distributions. PDM also generates a deviated distribution due to the frequent projection steps. Both FM-DD and FM-RE can have much higher probabilities of generating samples close enough to the subspace while maintaining the distributional similarity with the training distribution. In addition to the stats in Table 2, we report average distances between the generated samples and the subspace: $14.78 \times 10^{-4}$ (FM), $2.08 \times 10^{-4}$ (FM-DD), and $2.32 \times 10^{-4}$ (FM-RE), which also illustrate that the proposed methods can generate samples closer to the subspace.

## 6.2 MNIST Digits Generation with Certain Attributes

In this subsection, we consider the task of generating MNIST digits with the following constraints. We define pixels with values greater than 128 as white pixels. All other pixels are defined as black pixels.

**Brightness:** The brightness constraint requires an image to have at least 100 bright pixels.

**Thickness:** The maximum thickness of a digit is measured as the maximum distance from a white pixel to its nearest black pixel. The thickness constraint requires an image to have a maximum thickness that is strictly greater than 2 and strictly less than 3.

Both constraint sets are non-convex, and the distance functions to both constraint sets exist, but are discrete. A projection onto the brightness constraint set is available, whereas a projection onto the thickness constraint set is unavailable. These features determine that among the constraint-aware approaches, only FM-RE and PDM can apply to the **Brightness** case, and only FM-RE can apply to the **Thickness** case. Fig. 3 shows the generated images from FM and FM-RE, and all generated digits are clean and clearly recognizable. Table 3 shows the FID and constraint violation rate of the compared methods. For the brightness constraint, PDM generates samples with 0 constraint violation by design. FM-RE performs closely with FM in the FID metric, however, displays a much higher constraint satisfaction rate. Both constraints are non-convex and have unclear boundary information. The thickness constraint is an even more subtle requirement that may be

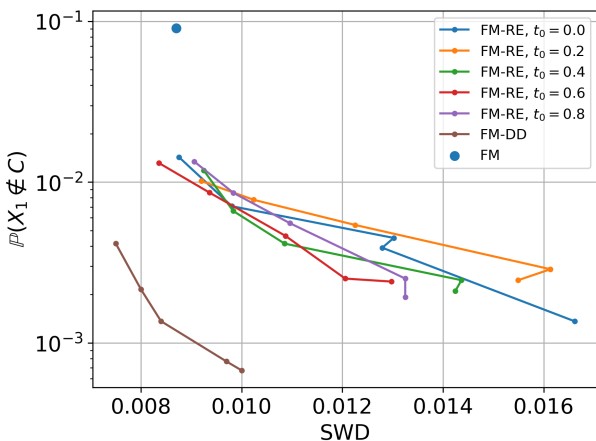

|  |  | Brightness | Thickness |
|---|---|---|---|
| FID ($\downarrow$) | FM | 6.16 | 6.19 |
|  | PDM | 8.06 | N/A |
|  | FM-RE | 5.86 | 10.8 |
| $\mathbb{P}(X_1 \notin \mathcal{C})$ ($\%, \downarrow$) | FM | 9.14 | 23.2 |
|  | PDM | 0 | N/A |
|  | FM-RE | 1.12 | 5.40 |

Table 3: Performance comparison for MNIST digits generation with certain attributes. We report the average $\mathbb{P}(X_1 \notin \mathcal{C})$ computed over 100 generation trials (1000 samples each). The Fréchet Inception Distance (FID) (Heusel et al., 2017) is computed based on a pre-trained LeNet-5 model between the training samples and $3 \times 10^4$ generated samples. Lower values indicate better performance for both metrics.

Figure 2: $\mathbb{P}(X_1 \notin C)$ vs. SWD for FM-RE with different $t_0$ and FM-DD by sweeping $\lambda$. Larger $\lambda$ generally corresponds with rightward motion.

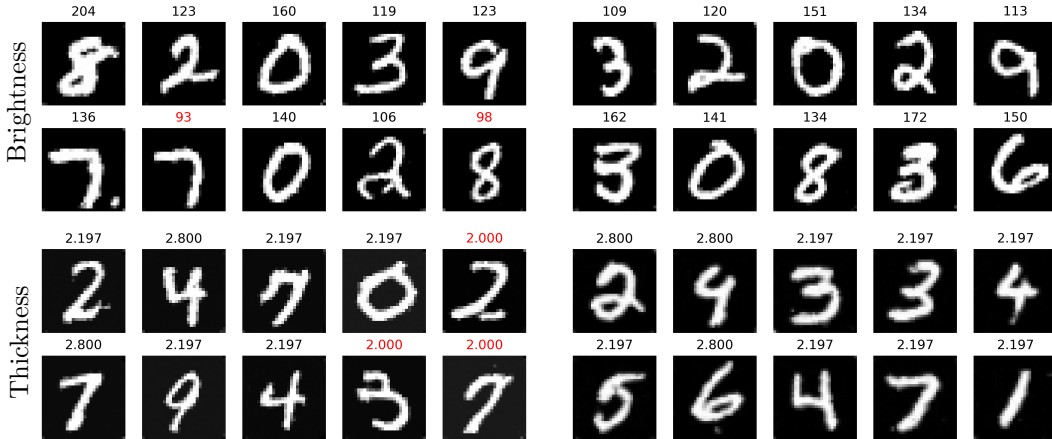

Figure 3: MNIST digits generated by different methods: **Left Panel**: FM, **Right Panel**: FM-RE. The number above each image denotes the number of white pixels / maximum thickness. Metrics violating the constraints are highlighted in red.

challenging for human observers to detect. Despite this, FM-RE is capable of generating samples satisfying constraints with a much higher probability than FM, demonstrating FM-RE's effectiveness.

Across all experiments, we conclude that FM-RE applies to the widest range of constraints and offers the advantage of maintaining a generated distribution close to the target distribution while exhibiting a low probability of generating samples violating the constraints.

## 6.3 Adversarial Example Generation for Hard-Label Black-Box Image Classification Models

We apply FM-RE to the task of training an adversarial example generator for hard-label black-box image classification models. The generated images are expected to be perceptually imperceptible to humans, while subject to the constraint that they must be assigned a label different from the ground truth. Notably, this requirement is a complex and unclear constraint, where membership is the only available information for the constraint.

Specifically, consider a black-box image classification model that takes in an image and outputs only the top-1 prediction's class label; the objective is to generate images whose true labels are different from the labels predicted by the classifier. The indicator function for the constraint is defined as follows

$$\mathbf{1}_{\mathcal{C}}\left(X_1^{\theta_2,\sigma}\right) = \begin{cases} 1, & \text{if} \quad \hat{y}\left(X_1^{\theta_2,\sigma}\right) \neq y\left(X_1^{\theta_2,\sigma}\right), \\ 0, & \text{otherwise.} \end{cases} \tag{13}$$

in which $X_1^{\theta_2,\sigma}$ is a generated sample. $y(\cdot)$ and $\hat{y}(\cdot)$ denote the ground truth and the classifier's label prediction, respectively. However, $y(\cdot)$ is hard to obtain for randomly generated samples unless manually checking them, especially when $X_1^{\theta_2,\sigma}$ are adversarial samples. To address this issue, we introduce the following rule: $X_{t_0}^{\theta_1} = (1-t_0)X_0 + t_0 X_1$. With $t_0$ set close to 1, the ground truth of $X_1^{\theta_2,\sigma}$ and $X_1$ are highly likely to be the same, i.e., $y\left(X_1^{\theta_2,\sigma}\right) = y(X_1)$. The objective is to generate a slightly perturbed adversarial version of $X_1$, i.e., $X_1^{\theta_2,\sigma}$. Next, we train adversarial example generators for two pre-trained models: LeNet-5 (Lecun et al., 1998) and ResNet-50 (He et al., 2016). To accommodate this task, in which the adversarial samples are initially unavailable, we adapt the clean training sets for FM-RE training.

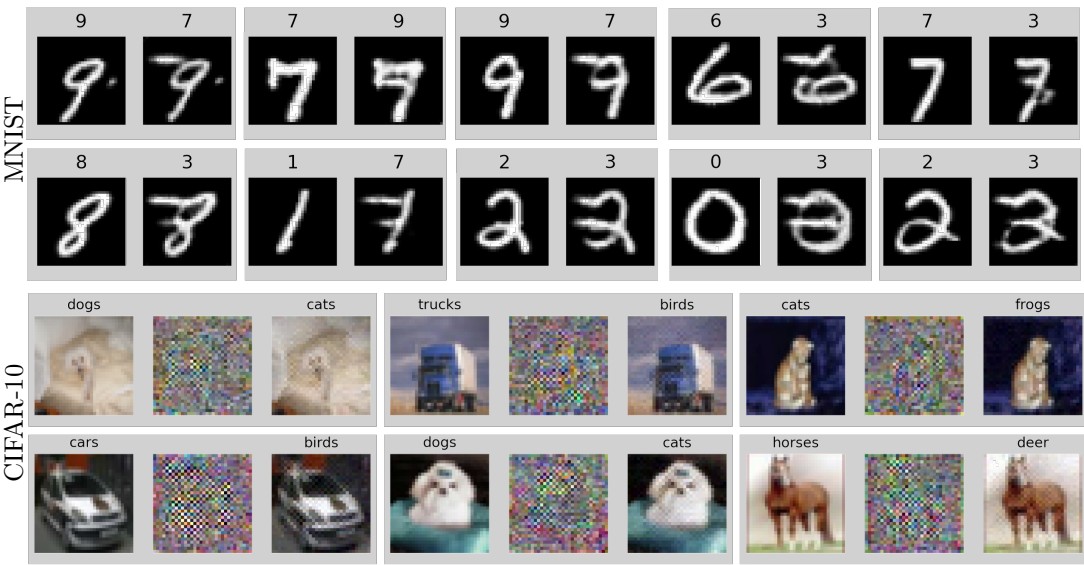

Figure 4: Pair-wise comparison between the clean images (left) and images generated via FM-RE (right). For CIFAR-10, we also provide the residue (middle). The predicted class by the corresponding pre-trained model is annotated above each image. The accuracy of LeNet-5 on MNIST (i.e., the constraint violation rate $\mathbb{P}(X_1 \notin \mathcal{C})$) drops from 99.1% to 18.7%. The accuracy of ResNet-50 on CIFAR-10 (i.e., the constraint violation rate $\mathbb{P}(X_1 \notin \mathcal{C})$) drops from 95.3% to 28.2%.

Fig. 4 shows the clean images along with the generated adversarial examples using FM-RE. We can observe that the FM-RE model is able to generate slightly perturbed images, which can misguide the classifier. A few images are perturbed to resemble a different digit (e.g., an '1' changed to a '7'), while others retain their original human-perceived class but are misclassified by the model. This further demonstrates FM-RE's capability to match the training distributions while adapting to complex constraints via membership querying.

## 6.4   Recommendations for Hyperparameter Selection

In practice, the key hyperparameter in FM-DD is the constraint weight $\lambda$, and the key hyperparameter in FM-RE includes the constraint weight $\lambda$, the randomization start time $t_0$, and the number of randomized steps $N_2$. In general, $\lambda$ controls the strength of constraint pressure. $t_0$ determines how early constraint-aware exploration influences the trajectory, and $N_2$ determines the sampling precisions. For novel tasks,

we recommend initializing the hyperparameters within the ranges $\lambda \in [10, 20]$, $t_0 \in [0.6, 0.8]$, and $N_2 = 20$, which in our experience provide a reasonable balance between constraint enforcement and sample fidelity. After obtaining a stable trained model, higher constraint satisfaction rates can be achieved by increasing $\lambda$ and decreasing $t_0$. If the constraint satisfaction rate is adequate but a degradation in sample quality is observed, one may instead decrease $\lambda$ and increase $t_0$ and/or $N_2$ to alleviate excessive constraint pressure while maintaining sufficient exploration capacity.

### 6.5 Limitations

The main limitations of this work are twofold. First, while FM-DD and FM-RE promote constraint satisfaction by penalizing constraint-violating samples, they do not provide formal guarantees on constraint satisfaction, and excessively large $\lambda$ may degrade the sample quality. Second, both methods require sampling full trajectories to evaluate terminal constraint satisfaction, leading to higher computational cost compared to conventional FM and certain alternative constrained generation approaches (e.g., MDM or reflection-based methods) whose objectives rely on single-step evaluations rather than complete trajectory simulations, as shown in Table 4. The additional computational overhead relative to conventional FM depends on the difficulty of the constraint: harder-to-satisfy constraints typically require more extensive exploration during training, thereby increasing the computational overhead.

|  | Box | 2 boxes | 8d $l_2$ ball | 20d $l_2$ ball | Subspace | MNIST brightness | MNIST thickness | LeNet-5 | CIFAR-10 |
|---|---|---|---|---|---|---|---|---|---|
| FM | 3 | 3 | 4 | 4 | 10 | 64 | 64 | N/A | N/A |
| RFM/MDM | 3 | 3 | 4 | 4 | 10 | N/A | N/A | N/A | N/A |
| NAMM | 8 | 8 | 12 | 12 | 24 | N/A | N/A | N/A | N/A |
| PDM | 3 | 3 | 4 | 4 | 10 | 64 | N/A | N/A | N/A |
| FM-DD | 5 | 5 | 8 | 10 | 20 | N/A | N/A | N/A | N/A |
| FM-RE | 5 | 5 | 8 | 10 | 20 | 85 | 192 | 712 | 2028 |

Table 4: Training time (in minutes) for each experiment.

## 7 Conclusions and Future Directions

In this paper, we present general constraint-aware FM frameworks: FM-DD and FM-RE. FM-DD incorporates a smooth, differentiable distance-based penalty for constraint violations, while FM-RE enforces constraint satisfaction by randomization directed to explore the constraints via access to a membership oracle. Compared to FM-RE, FM-DD yields empirically higher constraint satisfaction but requires access to a differentiable distance function to the constraint set. Thus, FM-DD is preferable when such distances are available, whereas FM-RE is suited to general settings with a membership oracle for the constraint set.

There are several pertinent research directions that this paper opens up: optimal choice of randomization time in FM-RE, quantification of the regret in using a randomized flow compared to the optimal constrained flow, provable guarantees on the likelihood of constraint satisfaction, and extension of the use of additional randomization in conjunction with other methods, such as DDPM, SDE-based, and Diffusion-Bridge based models.

## Acknowledgments

Zhengyan Huan and Shuchin Aeron are supported via NSF under Cooperative Agreement PHY-2019786 and DOE DE-SC0023964. Shuchin Aeron would also like to acknowledge funding via NSF DMS 2309519. Li-Ping Liu is supported by NSF Award 2239869.

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

## A Proof of Proposition 4.1

**Proposition.** *Let $X_1(\mathcal{T})$ denote the generated sample of trajectory $\mathcal{T}$, i.e., $X_1^{\theta,\sigma}$. Under assumptions of all $X_t^{\theta,\sigma}$ having strictly positive density on $\mathbb{R}^d$, $\mathcal{C}$ being a subset of $\mathbb{R}^d$ with positive Lebesgue measure, existence and uniqueness of the solutions driven by the respective ODEs, boundedness of the gradient of the log-densities, and finiteness of all the expectations involved:*

$$\nabla_{\theta,\sigma}\mathbb{E}\left[\mathbf{1}_{\mathcal{C}}(X_1^{\theta,\sigma})\right] = \mathbb{E}_{\mathcal{T}}\left[\sum_{i=0}^{N-1}\left(\nabla_{\theta,\sigma}\log\pi_{\theta,\sigma}(\tilde{U}_{i\Delta t}^{\theta,\sigma}|X_{i\Delta t}^{\theta,\sigma})\right)\mathbf{1}_{\mathcal{C}}(X_1(\mathcal{T}))\right].$$

*Proof.* Following equation 11, since by construction the variables $(X_{i\Delta t}, U_{i\Delta t})$ form a Markov chain and $(X_{(i-1)\Delta t}, U_{(i-1)\Delta t}) \perp U_{(i)\Delta t}|X_{(i)\Delta t}$, the probability density of a realization $\tau = (x_0^{\theta,\sigma}, \tilde{u}_0^{\theta,\sigma}, x_{\Delta t}^{\theta,\sigma}, \tilde{u}_{\Delta t}^{\theta,\sigma}, \cdots, x_1^{\theta,\sigma})$ of $\mathcal{T}$ is:

$$p(\tau;\theta,\sigma) = p(x_0^{\theta,\sigma})\prod_{i=0}^{N-1}p(x_{(i+1)\Delta t}^{\theta,\sigma}|x_{i\Delta t}^{\theta,\sigma},\tilde{u}_{i\Delta t}^{\theta,\sigma})\pi_{\theta,\sigma}(\tilde{u}_{i\Delta t}^{\theta,\sigma}|x_{i\Delta t}^{\theta,\sigma}). \tag{14}$$

Note that $x_0^{\theta,\sigma} \sim q_0$ and the transitions to $x_{(i+1)\Delta t}^{\theta,\sigma}$ given $x_{i\Delta t}^{\theta,\sigma}, \tilde{u}_{i\Delta t}^{\theta,\sigma}$ are independent of $\theta,\sigma$. Taking the gradient of the log probability density of $\tau$ leads to

$$\begin{aligned}
\nabla_{\theta,\sigma}\log p(\tau;\theta,\sigma) =& \nabla_{\theta,\sigma}\log p(x_0^{\theta,\sigma}) + \sum_{i=0}^{N-1}\nabla_{\theta,\sigma}\log p(x_{(i+1)\Delta t}^{\theta,\sigma}|x_{i\Delta t}^{\theta,\sigma},\tilde{u}_{i\Delta t}^{\theta,\sigma}) \\
& + \sum_{i=0}^{N-1}\nabla_{\theta,\sigma}\log\pi_{\theta,\sigma}(\tilde{u}_{i\Delta t}^{\theta,\sigma}|x_{i\Delta t}^{\theta,\sigma}) \\
=& \sum_{i=0}^{N-1}\nabla_{\theta,\sigma}\log\pi_{\theta,\sigma}(\tilde{u}_{i\Delta t}^{\theta,\sigma}|x_{i\Delta t}^{\theta,\sigma}).
\end{aligned} \tag{15}$$

Recall the objective in equation 7:

$$\begin{aligned}
\mathbb{E}\left[\mathbf{1}_{\mathcal{C}}(X_1^{\theta,\sigma})\right] =& \int p(x_1^{\theta,\sigma})\mathbf{1}_{\mathcal{C}}(x_1^{\theta,\sigma})dx_1^{\theta,\sigma} \\
=& \int p(x_0^{\theta,\sigma})\prod_{i=0}^{N-1}p(x_{(i+1)\Delta t}^{\theta,\sigma}|x_{i\Delta t}^{\theta,\sigma},\tilde{u}_{i\Delta t}^{\theta,\sigma})\pi_{\theta,\sigma}(\tilde{u}_{i\Delta t}^{\theta,\sigma}|x_{i\Delta t}^{\theta,\sigma})\cdot \\
& \mathbf{1}_{\mathcal{C}}(x_1^{\theta,\sigma})\ dx_1^{\theta,\sigma}d\tilde{u}_{1-\Delta t}^{\theta,\sigma}dx_{1-\Delta t}^{\theta,\sigma}\cdots d\tilde{u}_0^{\theta,\sigma}dx_0^{\theta,\sigma} \\
=& \int p(\tau;\theta,\sigma)\mathbf{1}_{\mathcal{C}}(x_1(\tau))\ d\tau.
\end{aligned} \tag{16}$$

Under the stated assumptions, by applying the Dominated Convergence Theorem, one can exchange the integration and differentiation, and we can write the gradient in equation 16 as:

$$\begin{aligned}
\nabla_{\theta,\sigma}\mathbb{E}\left[\mathbf{1}_{\mathcal{C}}(X_1^{\theta,\sigma})\right] =& \nabla_{\theta,\sigma}\int p(\tau;\theta,\sigma)\mathbf{1}_{\mathcal{C}}(x_1(\tau))\ d\tau \\
=& \int(\nabla_{\theta,\sigma}p(\tau;\theta,\sigma))\mathbf{1}_{\mathcal{C}}(x_1(\tau))\ d\tau & \text{Exchange integration and differentiation} \\
=& \int p(\tau;\theta,\sigma)(\nabla_{\theta,\sigma}\log p(\tau;\theta,\sigma))\mathbf{1}_{\mathcal{C}}(x_1(\tau))\ d\tau & \text{Log derivative trick} \\
=& \mathbb{E}_{\mathcal{T}}\left[(\nabla_{\theta,\sigma}\log p(\mathcal{T};\theta,\sigma))\mathbf{1}_{\mathcal{C}}(X_1(\mathcal{T}))\right] & \text{Definition of expectation} \\
=& \mathbb{E}_{\mathcal{T}}\left[\sum_{i=0}^{N-1}\left(\nabla_{\theta,\sigma}\log\pi_{\theta,\sigma}(\tilde{U}_{i\Delta t}^{\theta,\sigma}|X_{i\Delta t}^{\theta,\sigma})\right)\mathbf{1}_{\mathcal{C}}(X_1(\mathcal{T}))\right]. & \text{Plug in equation 15}
\end{aligned} \tag{17}$$

Note that in the formal derivation above we have assumed that all densities are strictly positive. $\qquad\square$

## B   Proof of Proposition 4.2

**Proposition.** *For randomization starting at time $t_0 > 0$, under the assumptions of Proposition 4.1 we have:*

$$
\nabla_{\theta,\sigma}\mathbb{E}\left[\mathbf{1}_{\mathcal{C}}(X_1^{\theta,\sigma})\right] = \mathbb{E}_{\mathcal{T}}[\nabla_{\theta,\sigma}\log p(X_{t_0}^{\theta})\mathbf{1}_{\mathcal{C}}(X_1(\mathcal{T}))] + \mathbb{E}_{\mathcal{T}}\left[\sum_{i=0}^{N_2-1}\left(\nabla_{\theta,\sigma}\log\pi_{\theta,\sigma}(\tilde{U}_{t_0+i\Delta t}^{\theta,\sigma}|X_{t_0+i\Delta t}^{\theta,\sigma})\right)\mathbf{1}_{\mathcal{C}}(X_1(\mathcal{T}))\right].
$$

*Proof.* The realization of a stochastic trajectory starts at $t = t_0$ and becomes $\tau = (x_{t_0}^{\theta,\sigma}, \tilde{u}_{t_0}^{\theta,\sigma}, x_{t_0+\Delta t}^{\theta,\sigma}, \tilde{u}_{t_0+\Delta t}^{\theta,\sigma}, \cdots, x_1^{\theta,\sigma})$ with $N_2$ steps. The PDF of $\tau$ is

$$
p(\tau;\theta,\sigma) = p(x_{t_0}^{\theta,\sigma})\prod_{i=0}^{N_2-1}p(x_{t_0+(i+1)\Delta t}^{\theta,\sigma}|x_{t_0+i\Delta t}^{\theta,\sigma}, \tilde{u}_{t_0+i\Delta t}^{\theta,\sigma})\pi_{\theta,\sigma}(\tilde{u}_{t_0+i\Delta t}^{\theta,\sigma}|x_{t_0+i\Delta t}^{\theta,\sigma}). \tag{18}
$$

It is important to mention that $x_{t_0}^{\theta} = x_{t_0}^{\theta,\sigma} = x_0^{\theta,\sigma} + \Delta t\sum_{i=0}^{N_1-1}u_\theta(x_{i\Delta t}^{\theta,\sigma}, i\Delta t)$ is dependent on $\theta$. This leads to $\nabla_{\theta,\sigma}\log p(x_{t_0}^{\theta,\sigma}) \neq 0$. Taking the gradient of the log probability density of $\tau$ gives

$$
\begin{aligned}
\nabla_{\theta,\sigma}\log p(\tau;\theta,\sigma) =& \nabla_{\theta,\sigma}\log p(x_{t_0}^{\theta,\sigma}) + \sum_{i=0}^{N_2-1}\nabla_{\theta,\sigma}\log p(x_{t_0+(i+1)\Delta t}^{\theta,\sigma}|x_{t_0+i\Delta t}^{\theta,\sigma}, \tilde{u}_{t_0+i\Delta t}^{\theta,\sigma})\\
&+ \sum_{i=0}^{N_2-1}\nabla_{\theta,\sigma}\log\pi_{\theta,\sigma}(\tilde{u}_{t_0+i\Delta t}^{\theta,\sigma}|x_{t_0+i\Delta t}^{\theta,\sigma})\\
=& \nabla_{\theta,\sigma}\log p(x_{t_0}^{\theta}) + \sum_{i=0}^{N_2-1}\nabla_{\theta,\sigma}\log\pi_{\theta,\sigma}(\tilde{u}_{t_0+i\Delta t}^{\theta,\sigma}|x_{t_0+i\Delta t}^{\theta,\sigma}),
\end{aligned} \tag{19}
$$

$$
\begin{aligned}
\mathbb{E}\left[\mathbf{1}_{\mathcal{C}}(X_1^{\theta,\sigma})\right] =& \int p(x_1^{\theta,\sigma})\mathbf{1}_{\mathcal{C}}(x_1^{\theta,\sigma})dx_1^{\theta,\sigma}\\
=& \int p(x_{t_0}^{\theta,\sigma})\prod_{i=0}^{N-1}p(x_{t_0+(i+1)\Delta t}^{\theta,\sigma}|x_{t_0+i\Delta t}^{\theta,\sigma}, \tilde{u}_{t_0+i\Delta t}^{\theta,\sigma})\pi_{\theta,\sigma}(\tilde{u}_{t_0+i\Delta t}^{\theta,\sigma}|x_{t_0+i\Delta t}^{\theta,\sigma})\cdot\\
&\mathbf{1}_{\mathcal{C}}(x_1^{\theta,\sigma})\ dx_1^{\theta,\sigma}d\tilde{u}_{1-\Delta t}^{\theta,\sigma}dx_{1-\Delta t}^{\theta,\sigma}\cdots d\tilde{u}_{t_0}^{\theta,\sigma}dx_{t_0}^{\theta,\sigma}\\
=& \int p(\tau;\theta,\sigma)\mathbf{1}_{\mathcal{C}}(x_1(\tau))\ d\tau.
\end{aligned} \tag{20}
$$

Under the assumption that one can exchange the integration and differentiation, and then by plugging in equation 19,

$$
\begin{aligned}
\nabla_{\theta,\sigma}\mathbb{E}\left[\mathbf{1}_{\mathcal{C}}(X_1^{\theta,\sigma})\right] =& \mathbb{E}_{\mathcal{T}}\left[(\nabla_{\theta,\sigma}\log p(\mathcal{T};\theta,\sigma))\mathbf{1}_{\mathcal{C}}(X_1(\mathcal{T}))\right]\\
=& \mathbb{E}_{\mathcal{T}}[\nabla_{\theta,\sigma}\log p(X_{t_0}^{\theta})\mathbf{1}_{\mathcal{C}}(X_1(\mathcal{T}))] + \mathbb{E}_{\mathcal{T}}\left[\sum_{i=0}^{N_2-1}\left(\nabla_{\theta,\sigma}\log\pi_{\theta,\sigma}(\tilde{U}_{t_0+i\Delta t}^{\theta,\sigma}|X_{t_0+i\Delta t}^{\theta,\sigma})\right)\mathbf{1}_{\mathcal{C}}(X_1(\mathcal{T}))\right].
\end{aligned} \tag{21}
$$

$\square$

We additionally note that, adding a baseline $b(X_{t_0+i\Delta t}^{\theta,\sigma})$ in equation 21 can preserve the unbiased gradient. For any fixed time step $i$,

$$
\begin{aligned}
&\mathbb{E}_{\mathcal{T}}\left[b(X_{t_0+i\Delta t}^{\theta,\sigma})\nabla_{\theta,\sigma}\log\pi_{\theta,\sigma}(\tilde{U}_{t_0+i\Delta t}^{\theta,\sigma}|X_{t_0+i\Delta t}^{\theta,\sigma})\right]\\
&=\mathbb{E}_{X_{t_0+i\Delta t}^{\theta,\sigma}}\left[b(X_{t_0+i\Delta t}^{\theta,\sigma})\mathbb{E}_{\tilde{U}_{t_0+i\Delta t}^{\theta,\sigma}\sim\pi_{\theta,\sigma}(\tilde{U}_{t_0+i\Delta t}^{\theta,\sigma}|X_{t_0+i\Delta t}^{\theta,\sigma})}\left[\nabla_{\theta,\sigma}\log\pi_{\theta,\sigma}(\tilde{U}_{t_0+i\Delta t}^{\theta,\sigma}|X_{t_0+i\Delta t}^{\theta,\sigma})\right]\right]\\
&=\mathbb{E}_{X_{t_0+i\Delta t}^{\theta,\sigma}}\left[b(X_{t_0+i\Delta t}^{\theta,\sigma})\int\pi_{\theta,\sigma}(\tilde{U}_{t_0+i\Delta t}^{\theta,\sigma}|X_{t_0+i\Delta t}^{\theta,\sigma})\nabla_{\theta,\sigma}\log\pi_{\theta,\sigma}(\tilde{U}_{t_0+i\Delta t}^{\theta,\sigma}|X_{t_0+i\Delta t}^{\theta,\sigma})\,d\,\tilde{U}_{t_0+i\Delta t}^{\theta,\sigma}\right]\\
&=\mathbb{E}_{X_{t_0+i\Delta t}^{\theta,\sigma}}\left[b(X_{t_0+i\Delta t}^{\theta,\sigma})\int\nabla_{\theta,\sigma}\pi_{\theta,\sigma}(\tilde{U}_{t_0+i\Delta t}^{\theta,\sigma}|X_{t_0+i\Delta t}^{\theta,\sigma})\,d\tilde{U}_{t_0+i\Delta t}^{\theta,\sigma}\right]\\
&=\mathbb{E}_{X_{t_0+i\Delta t}^{\theta,\sigma}}\left[b(X_{t_0+i\Delta t}^{\theta,\sigma})\nabla_{\theta,\sigma}\int\pi_{\theta,\sigma}(\tilde{U}_{t_0+i\Delta t}^{\theta,\sigma}|X_{t_0+i\Delta t}^{\theta,\sigma})\,d\tilde{U}_{t_0+i\Delta t}^{\theta,\sigma}\right]\\
&=\mathbb{E}_{X_{t_0+i\Delta t}^{\theta,\sigma}}\left[b(X_{t_0+i\Delta t}^{\theta,\sigma})\nabla_{\theta,\sigma}1\right]\\
&=0.
\end{aligned}
\tag{22}
$$

Therefore equation 21 can also be written as

$$
\begin{aligned}
&\nabla_{\theta,\sigma}\mathbb{E}\left[\mathbf{1}_{\mathcal{C}}(X_1^{\theta,\sigma})\right]\\
&=\mathbb{E}_{\mathcal{T}}[\nabla_{\theta,\sigma}\log p(X_{t_0}^{\theta})\mathbf{1}_{\mathcal{C}}(X_1(\mathcal{T}))]+\mathbb{E}_{\mathcal{T}}\left[\sum_{i=0}^{N_2-1}\left(\nabla_{\theta,\sigma}\log\pi_{\theta,\sigma}(\tilde{U}_{t_0+i\Delta t}^{\theta,\sigma}|X_{t_0+i\Delta t}^{\theta,\sigma})\right)\left(\mathbf{1}_{\mathcal{C}}(X_1(\mathcal{T}))-b(X_{t_0+i\Delta t}^{\theta,\sigma})\right)\right].
\end{aligned}
\tag{23}
$$

## C   Experiment Details

All experiments are run on an NVIDIA L40 GPU with 46 GB memory. The parameter settings for each experiment are given in Table 5. In terms of computational cost, the training stage of FM-DD and FM-RE requires more computation than FM since the full generation process needs to be simulated. The training time required for the proposed methods is also closely related to the complexity of the constraints. In general, more complicated constraints require more time to converge since more exploration is required for the model to learn how to satisfy them. FM-DD's and FM-RE's computational costs at the sampling stage are similar to that of FM. We report the training time for all experiments (in minutes) in Table 4. Notably, RFM, MDM, and PDM have total training times comparable to vanilla generative models (e.g., FM and DDPM) and incur only marginal additional training costs. For MDM, the computational overhead arises from computing the closed-form bijective projection between the constraint set and the unconstrained space, which typically requires minimal cost in applicable cases. For RFM, the overhead comes from computing reflections when trajectories reach constraint boundaries, which likewise introduces only minor computational cost. PDM is an inference-time method and does not modify the training stage of diffusion models. Similar to the proposed methods, NAMM also incurs additional training costs, primarily due to training the forward and backward projection models between the constraint set and the unconstrained space.

### C.1   Synthetic Experiments

The detailed results for synthetic experiments, including standard deviations, are presented in Table 6. Detailed descriptions of the compared baselines are provided as follows.

**FM (Lipman et al., 2023):**   Flow matching (FM) is a popular framework for generative models without considering requirements on constraint satisfactions.

**RFM (Xie et al., 2024):**   Reflected flow matching (RFM) extends FM to constrained domains by introducing reflected ODEs that keep trajectories inside the constraint boundaries. Explicit algorithmic

| | FM-DD | FM-RE | | | |
|---|---|---|---|---|---|
| | $\lambda$ | $N_1$ | $N_2$ | $t_0$ | $\lambda$ |
| Box (Sec. 6.1) | 80 | 60 | 15 | 0.8 | 80 |
| 2 boxes (Sec. 6.1) | 80 | 60 | 15 | 0.8 | 80 |
| 8d $\ell_2$ ball (Sec. 6.1) | 20 | 60 | 15 | 0.8 | 20 |
| 20d $\ell_2$ ball (Sec. 6.1) | 20 | 60 | 15 | 0.8 | 20 |
| Subspace (Sec. 6.1) | 1 | 60 | 10 | 0.9 | 1 |
| MNIST-brightness (Sec. 6.2) | N/A | 60 | 20 | 0.6 | 10 |
| MNIST-thickness (Sec. 6.2) | N/A | 60 | 20 | 0.6 | 10 |
| LeNet-5 (Sec. 6.3) | N/A | N/A | 15 | 0.8 | 6 |
| ResNet-50 (Sec. 6.3) | N/A | N/A | 15 | 0.8 | 20 |

Table 5: Parameter settings for each experiment.

designs for constraints, including reflected CNFs, analytical constraint-preserving conditional velocity fields, and a reflection-based sampling procedure, are required.

**MDM (Liu et al., 2023a):** Mirror diffusion models (MDM) propose learning diffusion models on convex constraint sets by mapping data to an unconstrained dual space via mirror maps and performing standard diffusion there. Explicit algorithmic designs for constraints, including analytic mirror maps and inverse mappings that guarantee constraint satisfaction by construction, are required.

**NAMM (Feng et al., 2025):** Neural approximate mirror map (NAMM) proposes learning approximate mirror maps and their inverses to transform constrained data into an unconstrained mirror space for diffusion modeling, handling general (possibly non-convex) constraints via differentiable distance functions. Explicit algorithmic designs for constraints, including the differentiable distance functions to the constraint sets and constraint-aware training losses, are required.

**PDM (Christopher et al., 2024):** Projected diffusion model (PDM) keep the training process of conventional diffusion models, and enforces constraint satisfaction by inserting projection steps onto arbitrary constraint sets at each sampling iteration. Explicit algorithmic designs for constraints, including a projected update and a projection operator onto constraint sets, are required.

| | | Box | 2 boxes | 8d $\ell_2$ ball | 20d $\ell_2$ ball | Subspace |
|---|---|---|---|---|---|---|
| SWD | FM | $0.1268 \pm 0.0692$ | $0.2260 \pm 0.0691$ | $0.0193 \pm 0.0029$ | $0.0087 \pm 0.0008$ | $0.0372 \pm 0.0039$ |
| | RFM | $0.1258 \pm 0.0688$ | N/A | $0.0177 \pm 0.0026$ | $0.0098 \pm 0.0009$ | N/A |
| | MDM | $0.1431 \pm 0.0747$ | N/A | $0.0292 \pm 0.0017$ | $0.0159 \pm 0.0044$ | N/A |
| | NAMM | $0.1680 \pm 0.0733$ | $0.2101 \pm 0.0795$ | $0.0228 \pm 0.0037$ | $0.0205 \pm 0.0008$ | $0.2124 \pm 0.0137$ |
| | PDM | $0.1268 \pm 0.0759$ | $0.5661 \pm 0.0409$ | $0.0200 \pm 0.0031$ | $0.0257 \pm 0.0018$ | $0.3079 \pm 0.0046$ |
| | FM-DD | $0.1228 \pm 0.0726$ | $0.2174 \pm 0.0809$ | $0.0175 \pm 0.0027$ | $0.0086 \pm 0.0008$ | $0.0356 \pm 0.0034$ |
| | FM-RE | $0.1250 \pm 0.0685$ | $0.2104 \pm 0.0783$ | $0.0194 \pm 0.0029$ | $0.0132 \pm 0.0010$ | $0.0355 \pm 0.0033$ |
| $\mathbb{P}(X_1 \notin \mathcal{C})$ (‰) | FM | $1.132 \pm 0.3444$ | $4.580 \pm 0.6012$ | $23.67 \pm 1.4322$ | $90.82 \pm 3.1783$ | $790.1 \pm 3.4733$ |
| | RFM | 0 | N/A | 0 | 0 | N/A |
| | MDM | 0 | N/A | 0 | 0 | N/A |
| | NAMM | $0.387 \pm 0.2011$ | $0.379 \pm 0.1868$ | $0.138 \pm 0.0792$ | $2.890 \pm 0.5281$ | $707.64 \pm 3.6507$ |
| | PDM | 0 | 0 | 0 | 0 | 0 |
| | FM-DD | $0.053 \pm 0.0768$ | $0.073 \pm 0.0968$ | $0.140 \pm 0.1114$ | $0.502 \pm 0.2433$ | $86.24 \pm 2.7056$ |
| | FM-RE | $0.066 \pm 0.0839$ | $0.222 \pm 0.1285$ | $0.768 \pm 0.2502$ | $2.513 \pm 0.5170$ | $98.58 \pm 2.9260$ |

Table 6: Performance comparison for synthetic experiments. The values stand for: mean $\pm$ std.

### C.1.1 2-D Toy Examples

**Box** The first case we consider is a cropped Gaussian distribution constrained by a box. The constraint set is given as

$$\mathcal{C} = \{(x_1, x_2) | -4 \leq x_1 \leq 4, -4 \leq x_2 \leq 4\}. \tag{24}$$

$q_1$ is the mixture of two Gaussians: $\mathcal{N}\left(\begin{bmatrix} 3 \\ 3 \end{bmatrix}, \begin{bmatrix} 0.6 & 0 \\ 0 & 0.6 \end{bmatrix}\right)$ and $\mathcal{N}\left(\begin{bmatrix} -3 \\ -3 \end{bmatrix}, \begin{bmatrix} 1.5 & 0 \\ 0 & 1.5 \end{bmatrix}\right)$ with equal mixing weights, which are truncated by $\mathcal{C}$.

**2 boxes** The second case is a uniform distribution on two disconnected boxes. The constraint set is

$$\mathcal{C} = \{(x_1, x_2) | 1 \leq |x_1| \leq 5, 1 \leq |x_2| \leq 5, x_1 x_2 > 0\}. \tag{25}$$

The target distribution is the uniform distribution in $\mathcal{C}$.

**FM-RE vs. Fine-Tuning:** We contrast our proposed joint optimization methods with the idea of fine-tuning methods (Uehara et al., 2024) in the 2 boxes case. Uehara et al. (2024) proposes to involve RL-based fine-tuning in diffusion models to optimize a *new* downstream objective while preserving proximity to the base model's output. Following this idea, a fine-tuning objective can be defined via

$$\arg\min_\theta \int_0^1 \mathbb{E}[\|u_\theta(X_t, t) - u_{\text{base}}(X_t, t)\|^2]dt + \lambda \, \mathbb{E}[d(X_1^\theta, \mathcal{C})], \tag{26}$$

in which $u_{\text{base}}$ is a pre-trained FM model and $u_{\text{base}}$ is the fine-tuned model. Note that fine-tuning is also a two-stage approach, however, the purposes are obtaining a base model and updating it for another task. FM-RE's two stages serve for different purposes, i.e., obtaining the estimated velocity field for $t < t_0$ and $t \geq t_0$, respectively.

The comparison between FM-RE and a fine-tuned FM model is shown in Fig. 5. We can observe a clear distribution shift in a fine-tuned model, i.e., the white gap near the boundary. The results also show a larger SWD for the fine-tuned model, while the constraint satisfaction rate for both models remains close. It is important to note that the two main objectives in constraint generation, i.e., matching distributions and satisfying constraints, are well aligned. In such cases, joint optimization is preferable to fine-tuning, as it enables the model to learn a shared representation that simultaneously achieves both objectives. The proposed joint optimization method, FM-RE, avoids the risk of forgetting to learn the distribution in fine-tuning, while also providing a more balanced trade-off between the two objectives.

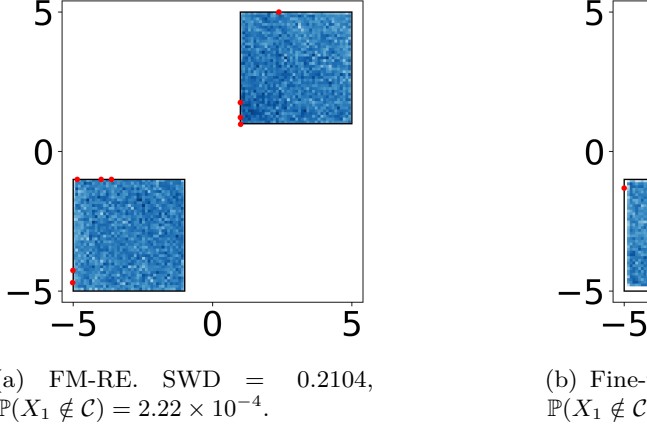

(a) FM-RE. SWD $=$ 0.2104, $\mathbb{P}(X_1 \notin \mathcal{C}) = 2.22 \times 10^{-4}$.

(b) Fine-tuning. SWD $= 0.2755$, $\mathbb{P}(X_1 \notin \mathcal{C}) = 2.46 \times 10^{-4}$.

Figure 5: The histplots of samples generated by FM-RE and a fine-tuned FM model. Samples violating the constraints are highlighted in red. The total sample size is 50000.

### C.1.2 Gaussian Mixture Distribution with $\ell_2$ Ball Constraints

We next evaluate the proposed methods for distributions and constraints with higher dimensions following Liu et al. (2023a). The constraint is an $\ell_2$ ball constraint, which can be given as

$$\mathcal{C} = \left\{ x \in \mathbb{R}^d \middle| \|x\|_2 \leq 1 \right\}, \tag{27}$$

in which the dimension is selected to be $d = \{8, 20\}$. The target distribution $q_1$ is a Gaussian mixture model. We consider $d$ isotropic Gaussians, each with variance 0.05, centered at each of the $d$ standard unit vectors, and reject samples outside $\mathcal{C}$.

In this case, we additionally report and analyze the mean gradient norm and variance during the training process. Following equation 23, adding a baseline $b(X^{\theta,\sigma}_{t_0+i\Delta t})$ preserves an unbiased estimation of the gradient. Subtracting a baseline centered around the mean of the indicator term can reduce the magnitude of stochastic fluctuations, thereby reducing the gradient variance. While one may choose to train an additional neural network to model the baseline, we provide an efficient choice of setting $b(X^{\theta,\sigma}_{t_0+i\Delta t}) = 1$. As we are expecting most of the generated samples to satisfy the constraints, $\mathbf{1}_{\mathcal{C}}(X_1(\mathcal{T})) - b(X^{\theta,\sigma}_{t_0+i\Delta t})$ becomes 0 for most of the trajectories, especially for the end of the training process. The training processes of two models—one with a baseline and one without—are shown in Fig. 6. The results show that adding a baseline $b(X^{\theta,\sigma}_{t_0+i\Delta t}) = 1$ effectively reduces the mean gradient norm and gradient variance during the training process. At the end of the training process, both models achieve similar performance as shown in Table 2.

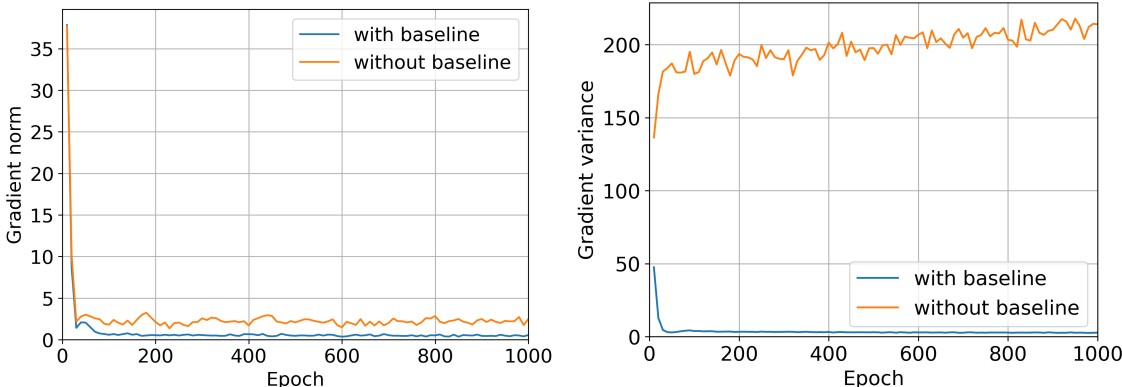

Figure 6: The mean gradient norm and variance during the training process. At each epoch, the model parameters are fixed, and the same training epoch is repeated 100 times to obtain 100 gradient vectors. The mean gradient norm and variance are then computed from these 100 gradients. "with baseline" follows the objective in equation 23 with $b(X^{\theta,\sigma}_{t_0+i\Delta t}) = 1$ and "without baseline" follows the objective in equation 21.

### C.1.3 Subspace Constraint

The considered subspace constraint is given by

$$\mathcal{C} = \left\{ x \in \mathbb{R}^{10} | \mathbf{1}_{10}^{\top} x + 10 = 0 \right\}. \tag{28}$$

The target distribution $q_1$ is the projection of a 10-D Gaussian distribution to a subspace. Specifically, we first generate a 10-D multivariate Gaussian distribution. The samples are denoted by $x \in \mathbb{R}^{10}$ with $x \sim \mathcal{N}(\mu, \Sigma)$, where $\mu = \mathbf{0}_{10} \in \mathbb{R}^{10}$ and $\Sigma = \boldsymbol{I}_{10}$. Then each sample is orthogonally projected to a 9-D hyperplane $\mathcal{C}$. The generated samples are not likely to strictly belong to $\mathcal{C}$ since it has no interior. Therefore, a small error is allowed and the indicator function is redefined as

$$\mathbf{1}_{\mathcal{C}}(x) = \begin{cases} 1, & \text{if } d(x, \mathcal{C}) \leq 5 \times 10^{-4}, \\ 0, & \text{otherwise.} \end{cases} \tag{29}$$

## C.2 MNIST Digits Generation with Certain Attributes

### C.2.1 Brightness Constraint

We first specify a brightness constraint. For each MNIST image, we create a binary version of it by applying a threshold of 128: pixel values greater than 128 are set to 255 (white), and all others are set to 0 (black).

| $\lambda$ | 100 | 50 | 30 | 20 | 10 | 5 | 2 | 1 | 0.5 | 0.1 | 0.01 | 0 (FM) |
|---|---|---|---|---|---|---|---|---|---|---|---|---|
| $\mathbb{P}(X_1 \notin \mathcal{C})(\%,\downarrow)$ | 0.78 | 0.89 | 0.65 | 0.96 | 1.12 | 1.36 | 1.34 | 1.51 | 3.09 | 6.46 | 8.56 | 9.14 |
| FID ($\downarrow$) | 10.47 | 7.91 | 7.13 | 6.38 | 5.86 | 6.17 | 5.51 | 5.89 | 5.69 | 5.97 | 6.23 | 6.16 |

Table 7: FM-RE's trade-off between constraint violation rate, sample quality (FID) and $\lambda$. $t_0 = 0.6$ is fixed.

| $t_0$ | 0 | 0.1 | 0.2 | 0.3 | 0.4 | 0.5 | 0.6 | 0.7 | 0.8 | 0.9 | 1 (FM) |
|---|---|---|---|---|---|---|---|---|---|---|---|
| $\mathbb{P}(X_1 \notin \mathcal{C})(\%,\downarrow)$ | 0.42 | 0.02 | 0.17 | 0.10 | 0.24 | 0.66 | 1.12 | 2.83 | 4.63 | 6.92 | 9.14 |
| FID ($\downarrow$) | 10.41 | 21.96 | 13.27 | 16.43 | 9.52 | 8.82 | 5.86 | 5.66 | 6.28 | 5.99 | 6.16 |

Table 8: FM-RE's trade-off between constraint violation rate, sample quality (FID) and $t_0$. $\lambda = 10$ is fixed.

An image is considered bright if its binary version contains at least 100 white pixels (i.e., pixels with value 255); otherwise, it is considered dark. The objective is to generate only bright images based on this rule.

Out of the MNIST training set, 30379 images are bright images. We select this subset of bright images as our training set. It is important to note that both the training images and the generated images remain in their original (non-binary) form. Brightness constraints on the clean images are based solely on their binary versions.

To evaluate the performance of the generated samples, we compute the Fréchet Inception Distance (FID) between generated digits and real digits in the training set. Since MNIST images are grayscale and have different features compared with natural images, we use a LeNet-5 model pre-trained on MNIST as the feature extractor. Specifically, we remove the final classification layer of LeNet-5 and extract features for both generated and real images. The FID is then computed based on the mean and covariance of these features.

The following experiment aims for a further empirical study on FM-RE's trade-off between constraint satisfaction and sample quality by the choice of key hyperparameters $\lambda$ and $t_0$. First $t_0 = 0.6$ is fixed, and $\lambda$ varies from 100 to 0. Then $\lambda = 0.6$ is fixed, and $t_0$ varies from 0 to 1. Note that either setting $\lambda = 0$ or setting $t_0 = 1$ makes FM-RE equivalent to FM. The results are shown in Table 7 and Table 8, respectively. We can observe that both fixing $t_0$, increasing $\lambda$ and fixing $\lambda$, moving $t_0$ towards 0 generally increases the constraint satisfaction rate and decreases the sample quality. If both $t_0$ and $\lambda$ are set properly, FM-RE's sample quality will be similar to FM's; however, FM-RE's constraint satisfaction rate is much higher than FM's.

### C.2.2 Maximum Thickness Range Constraint

Secondly, we consider a maximum thickness range constraint, which is also based on the binary version of each image. The maximum thickness of a digit is measured as the maximum distance from a white pixel to its nearest black pixel. It is measured via `cv2.distanceTransform` function in the implementation. We define an image to satisfy this constraint if its maximum thickness is strictly greater than 2 and strictly less than 3. 21405 images in the MNIST training set satisfy this constraint, and the subset consisting of them is chosen as the training set for this task.

### C.3 Adversarial Example Generation for Hard-Label Black-Box Image Classification Models

### C.3.1 LeNet-5 Model for MNIST Digits Classification

LeNet-5 (Lecun et al., 1998) is a classic CNN architecture for MNIST digits classification, which accepts $32 \times 32$ grayscale images. Note that MNIST consists of $28 \times 28$ images. We first resize every image in MNIST to $32 \times 32$. The pretrained model has an accuracy of 99.1%. The training set of this pre-trained model should not be accessible due to the black-box setting. We split the testing set of MNIST into our training and testing sets using a 80 : 20 ratio.

For evaluation on our testing set, we only generate once for each image. Table 9 shows that The average $\ell_2$ norm between the generated images and the clean images is 5.47 (4.19 if scaled back to $28 \times 28$). Adding a similar level of Gaussian noise has almost no influence on LeNet-5's accuracy. However, samples from FM-RE can reduce the accuracy to 18.7%. The comparison between the clean images and the generated samples

|  | $\ell_2$ norm | | | Accuracy (%) | | |
|---|---|---|---|---|---|---|
|  | Clean | Gaussian | FM-RE | Clean | Gaussian | FM-RE |
| MNIST | 0 | 5.47 | 5.47 | 99.1 | 99.2 | 18.7 |
| CIFAR-10 | 0 | 11.15 | 11.15 | 95.3 | 82.3 | 28.2 |

Table 9: Performance comparison among clean samples, samples with Gaussian noise (added to a similar level of $\ell_2$ norm with FM-RE), and samples generated by FM-RE.

is shown in Fig. 7. We can observe that the FM-RE model is able to modify the clean digits in order to misguide the classifier. A few images are perturbed to resemble a different digit (e.g., an '1' changed to a '7'), while others retain their original human-perceived digit but are misclassified by the model.

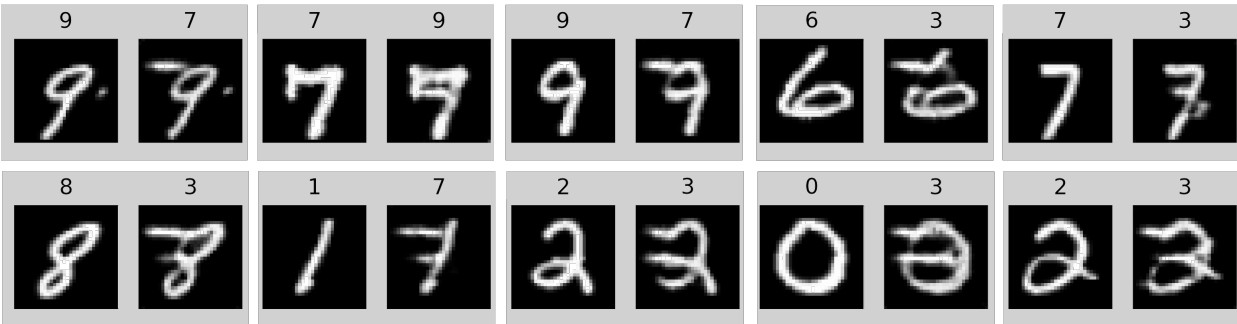

Figure 7: The comparison between the clean images and generated adversarial examples. For each image pair, the left is the clean image and the right is the generated adversarial example. The predicted class by the pre-trained LeNet-5 is annotated above each image.

### C.3.2 ResNet-50 Model for CIFAR-10 Images Classification

ResNet-50 (He et al., 2016) is a widely-used deep CNN structure with residual connections. The pre-trained model's accuracy on CIFAR-10 is 95.3%. We also split the testing set of CIFAR-10 into our training and testing set using an 80 : 20 ratio.

FM-RE is evaluated in the same way as the evaluation approach for LeNet-5. As shown in Table 9, PG-FM can significantly impact the accuracy of ResNet-50. We can also observe the samples shown in Fig. 8 that generated images can be recognized by humans, however cannot be correctly classified by ResNet-50.

Although the main purpose of this case is to illustrate FM-RE's adaptability to complex constraints, it is worth mentioning that images generated by FM-RE usually have a larger $\ell_2$ norm w.r.t. the clean images compared to state-of-the-art hard-label black-box adversarial example generation methods (Cheng et al., 2020; Park et al., 2024; Brendel et al., 2018). The reason is that FM-RE's objective does not include reducing $\ell_2$ norm. Instead, it seeks to regenerate the image using a generative model, which inherently creates new patterns. Although new patterns are introduced, the generated images remain visually similar to the originals, as shown in both cases. Moreover, FM-RE has the following advantages,

1. A common strategy for identifying potential adversarial example queries is by checking repeated queries for similar images. This might not be effective against FM-RE. FM-RE's training requires diverse queries, provided that the selected $x_1$ are sufficiently different.

2. FM-RE requires no query access when generating images, leading to fast generation. Also, one can generate an infinite number of potential adversarial examples for a single image by repeatedly sampling $x_0$.

The first advantage of FM-RE provides additional insight into the protection of image classification models. A common and effective defense is to block the sender when receiving repeated similar queries, however, this

is not enough since diverse queries may also be used to train an adversarial example generator. Monitoring unexpected features in the queried images can potentially identify queries that aim to create adversarial examples.

In conclusion, the adversarial example generation task illustrates that FM-RE is capable of adapting to complex and unclear constraints, even when the constraints only provide the information of membership.

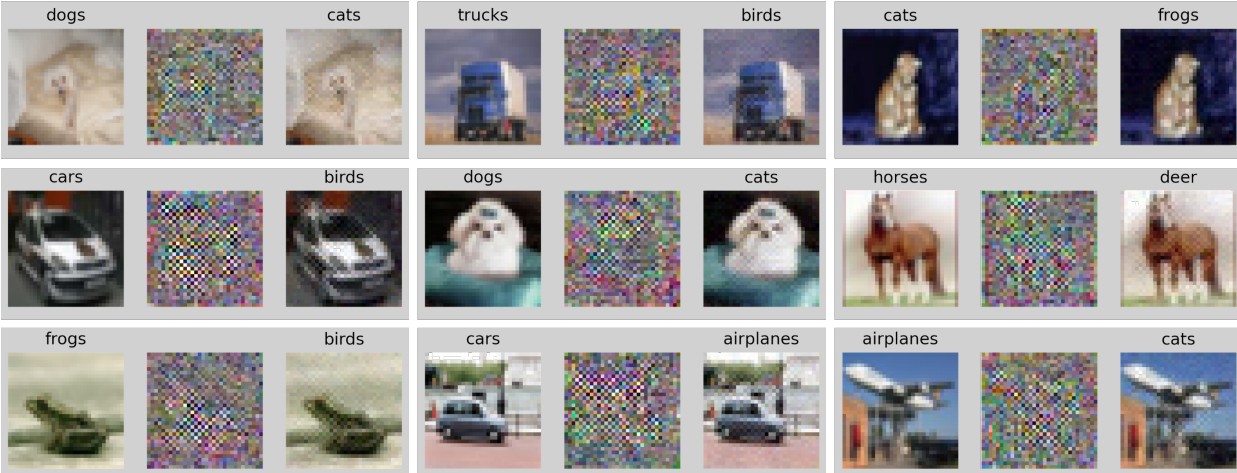

Figure 8: The comparison between the clean images and generated adversarial examples. For each image pair, the left is the clean image and the right is the generated adversarial example. The middle is the amplified perturbation between the generated images and the clean images. The predicted class by the pre-trained ResNet-50 is annotated above each image.

