# OpenReview forum: "Constraint-Aware Flow Matching via Randomized Exploration"
_TMLR — Accepted by TMLR_

### Review · Reviewer_1z3G · 2026-02-21

**Summary Of Contributions:**

This paper proposes a flow matching objective that explicitly accommodates scenarios where the target data distribution is supported on a constrained set. The authors consider two primary problem settings: (a) when a differentiable distance function to the constraint set is available, and (b) when the constraints can only be probed via a membership oracle. To address setting (a), the authors append a penalty term to the flow matching objective corresponding to the distance between the generated samples and the constraint set. To address setting (b), the authors use a randomized stochastic velocity field to explore the constraint boundaries and learn a mean flow that is highly likely to satisfy the membership oracle. The paper presents experiments validating these approaches on synthetic datasets (e.g., l^2 balls and subspace constraints), as well as practical tasks like constrained MNIST generation and black-box adversarial example generation. The empirical results demonstrate that the proposed methods achieve significantly higher rates of constraint satisfaction compared to standard flow matching and other baselines, while largely preserving the quality of the generated distribution.

**Audience:**

Yes

**Audience Explanation:**

This work offers a highly pragmatic approach to an important problem and should be useful for researchers and practitioners.

**Broader Impact Concerns:**

None.

**Claims And Evidence:**

Yes

**Claims Explanation:**

The authors perform experiments in both synthetic and more realistic settings.

**Requested Changes:**

The authors acknowledge that explicitly satisfying constraints heavily relies on the penalty multiplier $\lambda$, and that an excessively large $\lambda$ degrades the distributional match. Furthermore, the performance and computational cost of FM-RE depend highly on the choice of the threshold time $t_0$ and the number of integration steps $N_1$ and $N_2$. The authors should include a more structured heuristic or rule-of-thumb in the main text to guide practitioners in selecting $\lambda$ and $t_0$ for novel datasets.

The paper notes in the limitations that training FM-DD and FM-RE requires sampling complete trajectories, which incurs a higher computational cost than standard FM. It would be beneficial to expand on this limitation in the main text, specifically discussing how this full-trajectory simulation scales when working with high-resolution image spaces or much larger network architectures.

---

> ### Author Response · Authors · 2026-03-06
>
> We sincerely thank you for carefully reading our manuscript and the insightful comments, which have greatly helped improve the quality and clarity of the paper. Please refer to the revised manuscript for the updates.
>
> Responses to the requested changes
>
>    1. We added a new Sec 6.4:
>
> “*In practice, the key hyperparameter in FM-DD is the constraint weight $\lambda$, and the key hyperparameter in FM-RE includes the constraint weight $\lambda$, the randomization start time $t_0$, and the number of randomized steps $N_2$. In general, $\lambda$ controls the strength of constraint pressure. $t_0$ determines how early constraint-aware exploration influences the trajectory, and $N_2$ determines the sampling precisions. For novel tasks, we recommend initializing the hyperparameters within the ranges  $\lambda \in [10, 20]$, $t_0 \in [0.6, 0.8]$, and $N_2 = 20$, which in our experience provide a reasonable balance between constraint enforcement and sample fidelity. After obtaining a stable trained model, higher constraint satisfaction rates can typically be achieved by increasing $\lambda$ and decreasing $t_0$. If the constraint satisfaction rate is adequate but a degradation in sample quality is observed, one may instead decrease $\lambda$ and increase $t_0$ and/or $N_2$ to alleviate excessive constraint pressure while maintaining sufficient exploration capacity.*”
>
>    2. We need to clarify a point that the **computational overhead relative to conventional FM** does not scale with the larger network structures or more complex images, instead, it scales with the difficulty of the constraint satisfaction.
>
>    This is shown empirically in the experiments. Heuristically, one may  say that a constraint is hard to satisfy if the conventional FM model achieves a low constraint satisfaction rate. In Sec. 6.2 and Sec 6.3, MNIST brightness, MNIST thickness and adversarial sample generator for LeNet tasks all require generating MNIST digits. As shown in Table 4, FM’s training time remains similar for these tasks, however, FM-RE’s training time varies. Then the easiest task (MNIST brightness) has the least computational overhead and the hardest task (adversarial sample generator for LeNet) has the largest computational overhead.
>
>    To clarify this, we add a discussion in Sec 6.5:
>
>    “*The main limitations of this work are twofold. First, while FM-DD and FM-RE promote constraint satisfaction by penalizing constraint-violating samples, they do not provide formal guarantees on constraint satisfaction, and excessively large $\lambda$ may degrade the sample quality. Second, both methods require sampling full trajectories to evaluate terminal constraint satisfaction, leading to higher computational cost compared to conventional FM and certain alternative constrained generation approaches (e.g., MDM or reflection-based methods) whose objectives rely on single-step evaluations rather than complete trajectory simulations, as shown in Table 4. The additional computational overhead relative to conventional FM depends on the difficulty of the constraint: harder-to-satisfy constraints typically require more extensive exploration during training, thereby increasing the computational overhead.*”

---

### Review · Reviewer_REGf · 2026-03-03

**Summary Of Contributions:**

This paper proposes two methods for constraint-aware generative modeling within the Flow Matching (FM) framework:
FM-DD: It adds a differentiable penalty term based on the distance to a constraint set when such a distance function is available.

FM-RE: It handles general constraints accessible only via a membership oracle by introducing stochastic randomization in the velocity field. It can construct an unbiased estimator of the objective, used as a tractable training objective.

Experiments are conducted on synthetic datasets and MNIST. The author also conducts experiments on the use of these methods in adversarial example generation.

**Audience:**

Yes

**Audience Explanation:**

Constrained generation with flow matching methods is an important research topic. Authors will be glad to see competitive empirical methods.

**Broader Impact Concerns:**

The paper aims to propose methods for constrained image generation. I do not find any potential impact that needs to be especially mentioned.

**Claims And Evidence:**

No

**Claims Explanation:**

The overall motivation and high-level claims of the paper are reasonable. However, the central idea, namely the introduction of randomized exploration, lacks sufficient conceptual clarification. While the subsequent derivations are technically straightforward and can be easily verified, the paper does not provide an intuitive or principled explanation of why injecting randomized velocity should improve constraint satisfaction. In particular, it remains unclear how the stochastic perturbation meaningfully alters the optimization landscape or increases the likelihood of satisfying the constraints.

Furthermore, as acknowledged in Appendix C, the proposed methods do not provide any explicit guarantees on constraint satisfaction. Even with the proposed method, the violation probability cannot be bounded. This limits the practical interpretability of the approach. For example, it remains unclear under what conditions FM-DD should be preferred over FM-RE, or vice versa; their relative effectiveness can only be determined empirically. The absence of theoretical guidance on this tradeoff represents an important limitation of the proposed framework.

**Requested Changes:**

1. More information about the introduction of the randomized exploration should be provided.

2. It should include a brief introduction of all the baselines used in the experiments, especially whether they contain any specific algorithm design for constraints.

3. The experiments are only conducted on small-scale datasets, like synthetic datasets and MNIST. Could you provide more results on some real-world image generation tasks, with the application of the proposed method on the state-of-the-art image generation models?

4. (minor) Could you provide more rigorous arguments on the effectiveness of the methods? It will not be a huge issue if it cannot be done.

---

> ### Author Response · Authors · 2026-03-06
>
> We sincerely thank you for carefully reading our manuscript and the insightful comments, which have greatly helped improve the quality and clarity of the paper. Please refer to the revised manuscript for the updates.
>
> Responses to the comments:
>
>    1. “The paper does not provide an intuitive or principled explanation of why injecting randomized velocity should improve constraint satisfaction. In particular, it remains unclear how the stochastic perturbation meaningfully alters the optimization landscape or increases the likelihood of satisfying the constraints.”
>
>    The primary motivation for introducing randomness is to enable the computation of $\nabla_\theta \mathbb{E}\_{\mathcal{C}}(X_1^\theta)$. As discussed at the beginning of Sec. 3, the likelihood of satisfying the constraints increases because the additional term $-\lambda\mathbb{E}\_{\mathcal{C}}(X_1^\theta)$ in Eq. (3) penalizes generating samples that violate the constraints. However, for a conventional FM model, $X\_1^\theta$ is deterministic provided with the initial point $X\_0$. Consequently, the gradient of this term is zero almost everywhere (except on the boundary), and this makes the computation of $\nabla_\theta \mathbb{E}\_{\mathcal{C}}(X_1^\theta)$ infeasible. By introducing randomness, the probability of a sampling trajectory becomes well-defined, thereby enabling gradient computation and making objective (3) tractable.
>
>    We make the following changes at the start of Sec. 4.2 to clarify this:
>
>    From “*Note that the gradient of an indicator function is $0$ almost everywhere except for the boundary. This poses challenges in solving for equation 3 via gradient descent-like methods.*”
>
>    To “*In a conventional FM model, the output $X_1^\theta$ is deterministic given the initial point $X_0$. As a result, the constraint term $\mathbb{E} [\mathbf{1}\_{\mathcal{C}}(X^{\theta}_{1})]$ reduces to an indicator function evaluated at a deterministic point. Since the gradient of an indicator function is $0$ almost everywhere except on the boundary, the gradient with respect to
> $\theta$ vanishes almost everywhere. One way to address this issue is to introduce randomness into the sampling process, so that $X_1^\theta$
>  becomes a random variable rather than a deterministic output. The constraint term then corresponds to a probability of satisfaction, yielding a differentiable expectation with respect to
> $\theta$ and enabling gradient-based optimization.*”
>
>    2. “Furthermore, as acknowledged in Appendix C, the proposed methods do not provide any explicit guarantees on constraint satisfaction. Even with the proposed method, the violation probability cannot be bounded. This limits the practical interpretability of the approach.”
>
>    We need to first clarify that a conventional FM model can theoretically achieve nearly $0$-constraint violation rate even without constraint-aware modification since the target distribution’s support is on $\mathcal{C}$. However, this is not likely to happen in real implementation due to the limitation in (a) capacity of the neural network, (b) number of training samples, (c) noise in the training process. The practical interpretability of the proposed approach lies in directly informing the model of the constraints, thus increasing the constraint satisfaction rate.
>
>    We believe it is possible to theoretically analyze FM-RE for guarantees on constraint satisfaction, possibly in terms of (a) the number of samples available for training, (b) how many querying (number of iterations) is needed for constraint exploration in the second stage, and (c) errors in the velocity estimation. Our initial thought is that this problem can be treated within the framework of learning with limited samples, noise and (limited) feedback setting, where each time we get a 0/1 signal about the constraint set while trying to match the distribution. This, to the best of our knowledge, is a novel problem and is substantial enough in its own right to be currently beyond the scope of the present submission.

---

> ### Author Response · Authors · 2026-03-06
>
> 3. “For example, it remains unclear under what conditions FM-DD should be preferred over FM-RE, or vice versa; their relative effectiveness can only be determined empirically. The absence of theoretical guidance on this tradeoff represents an important limitation of the proposed framework.”
>
>    Compared with FM-RE, FM-DD’s advantages lie on: (1) FM-DD does not need to introduce randomness in the sampling trajectory. (2) FM-DD’s gradient update is more stable since it optimizes a smooth, differentiable distance penalty. (3) FM-DD empirically achieves better constraint satisfaction performance than FM-RE, as shown in Table 2.
>
>    While the limitation of FM-DD compared with FM-RE is that FM-DD can only apply to the cases, in which a differentiable distance to the constraint set is available. For example, FM-DD cannot apply to the tasks in Sec, 6.2 and 6.3
>
>    In conclusion, we can choose between FM-DD and FM-RE based on the following principle: If a differentiable distance to the constraint set is available and easy to estimate, FM-DD is preferred. If the membership oracle to the constraint set is available and the differentiable distance is not available, FM-DD is not applicable and FM-RE is preferred.
>
>    We made the following changes in Sec. 7,
>
>    “In this paper, we present general constraint-aware FM frameworks: FM-DD and FM-RE. FM-DD incorporates a smooth, differentiable distance-based penalty for constraint violations, while FM-RE enforces constraint satisfaction by randomization directed to explore the constraints via access to a membership oracle. Compared to FM-RE, FM-DD yields empirically higher constraint satisfaction but requires access to a differentiable distance function to the constraint set. Thus, FM-DD is preferable when such distances are available, whereas FM-RE is suited to general settings with a membership oracle for the constraint set.”
>
> Responses to the requested changes
>
>    1. We refer the reviewer to Response to the comments 1.
>
>    2. We added the introduction of all the baselines with focus on  specific algorithm design for constraints in appendix D.1. We refer the reviewer to the updated manuscript for this introduction.
>
>    3. Our work focuses on proposing a general framework for constrained generation at the methodological level, rather than targeting large-scale real-world image generation specifically. The synthetic experiments and MNIST studies are designed to provide controlled and interpretable validation of the proposed methods. Importantly, beyond MNIST, we also evaluate a substantially more complex and higher-dimensional task—adversarial example generation against hard-label black-box classifiers (ResNet-50 on CIFAR-10), where the constraint is defined solely via a membership oracle and involves nontrivial semantic conditions (misclassification) rather than simple geometric structure (Sec. 6.3, Fig. 4). This setting is significantly more challenging than MNIST attribute control and demonstrates the applicability of our approach to realistic, oracle-based constraints. We want to maintain focus on the core methodological contribution.
>
>  Finally, while the proposed method is in principle compatible with large state-of-the-art image generators, it is currently not feasible in our setting due to computational resource constraints. Recent state-of-the-art image generation models (without constraint-aware modifications) typically require substantial GPU memory and compute for training or fine-tuning (often multiple high-memory GPUs, e.g., 8×A100 80GB or similar). Training state-of-the-art image generation models and FM-RE for new constrained datasets would therefore demand computational resources beyond those available to us. For this reason, we conduct experiments on smaller, widely used benchmark tasks that allow controlled evaluation while remaining computationally tractable.
>
>  4. We refer the reviewer to Response to the comments 3.

---

### Review · Reviewer_pKkV · 2026-03-10

**Summary Of Contributions:**

This paper proposes two methods for constraint-aware flow matching: FM-DD, which adds a differentiable distance penalty to the flow matching objective, and FM-RE, which introduces randomness into the flow to obtain a REINFORCE-style gradient through a binary membership oracle. The key contribution is FM-RE's generality. It requires only oracle access to constraint satisfaction, which is strictly weaker than all prior methods that need convexity, boundary normals, bijective projections, or differentiable distances. The theoretical development providing the gradient identity that enables FM-RE training is sound. The main weakness is a persistent gap between the strength of the claims and the supporting evidence, as detailed below.

**Audience:**

Yes

**Audience Explanation:**

Constrained generation is a practically relevant problem, and the oracle-only setting addressed by FM-RE is genuinely novel. The formalization of constraint-aware flow matching under two access models (differentiable distance vs. membership oracle) provides a useful framework. Several communities---e.g., those working on safety-constrained generation, scientific simulations with physical constraints, and adversarial robustness---would find the problem formulation and the theoretical machinery (REINFORCE through the flow) of interest, even if the current experimental evidence does not fully support the claims.

**Claims And Evidence:**

No

**Claims Explanation:**

* The abstract claims "significant gains in constraint satisfaction while matching the target distributions," yet FM-RE still violates constraints approximately 10% of the time in the highest-dimensional synthetic setting (20d L2​ ball, Table 2). The characterization of results is not accurate given non-trivial violation rates across all tested settings.

* The method is described as "computationally efficient," but Table 4 in the appendix shows FM-RE is up to 3x slower than FM.

* The REINFORCE gradient underpinning FM-RE is known to exhibit high variance with sparse binary rewards, yet variance is neither measured nor discussed.

* The SWD confidence intervals in Table 5 overlap between FM-DD/FM-RE and vanilla FM in multiple settings, casting doubt on whether some distributional matching improvements are statistically meaningful.

**Requested Changes:**

## Critical

1. Temper the language throughout to match the evidence. The abstract's "significant gains" and the conclusion's "enforces constraint satisfaction" are not supported by the results. FM-RE violates constraints ~10% of the time in the 20d setting.
2. Address the disconnect between motivation and experiments. The introduction cites watermark generation, fluid dynamics, and high-quality image generation as target applications, yet none appear in the experiments. Either include at least one such experiment or reframe the motivation around what is actually tested.
3. The adversarial example experiment sets $t_0 = 0.8$, so generation starts 80% of the way toward the training sample. This is tantamount to learned noise injection rather than constrained generation. The paper should clearly acknowledge this and report constraint satisfaction in the same $\mathbb{P}(X_1 \notin \mathcal{C})$ format used elsewhere, rather than only classifier accuracy drops, to enable consistent comparison across experiments.
4. Discuss when FM-RE should be preferred over FM-DD. FM-DD achieves lower violation rates in all shared settings (Table 2), and in most cases also matches or improves on FM-RE's SWD. Without this discussion, the contribution of FM-RE is unclear for settings where $d(\cdot, \mathcal{C})$ is available.

## Suggested

5. Report and discuss gradient variance of the REINFORCE estimator, or employ standard variance reduction techniques. The indicator function provides a sparse binary reward, and understanding the variance behavior is important for assessing practical stability beyond the small-scale settings tested.
6. Move computational cost comparisons (Table 4) into the main text. A claim of computational efficiency should be substantiated where it is made.

---

> ### Author Response · Authors · 2026-03-13
>
> We sincerely thank you for carefully reading our manuscript and the insightful comments, which have greatly helped improve the quality and clarity of the paper. All changes have been marked using colored edits to allow for quick review and verification.
>
> Responses to the comments:
>    1.  “FM-RE still violates constraints approximately 10% of the time in the highest-dimensional synthetic setting (20d L2​ ball, Table 2).”
>
>    Table 2 shows that FM-RE’s constraint violation rate is $2.513 ‰$, i.e., $0.2513$%, not $10$%. This might be caused by our usage of ‰ to avoid too many zeros, hope this clarifies the confusion. In this case, the constraint violation rate reduces from $90.82 ‰$ (conventional FM) to $2.513 ‰$ (FM-RE), and at the same time, FM-RE’s SWD to the training distribution is small (SWD = $0.0132$). Therefore, the statement "significant gains in constraint satisfaction while matching the target distributions" is reasonable.
>
>    2. “The method is described as "computationally efficient," but Table 4 in the appendix shows FM-RE is up to 3x slower than FM.”
>
> We will clarify that “computationally efficient” refers to two-stage FM-RE is more computationally efficient than the one-stage FM-RE. This is analyzed in Sec. 4.2 and validated in Sec 6.1 $\ell_2$ ball constraints: “*The training time required to complete the same number of iterations for $t_0 \in \{0,0.2,0.4,0.6,0.8\}$ is approximately $27:23:19:15:11$, respectively.*” Here $t_0=0$ represents the one-stage approach, while $t>0$ cases represents the two-stage approach.
>
>  To make this clear, we have made the following changes in the abstract (see the revised version),
>
> “*Furthermore, in the proposed setting we show that a two-stage approach, where both stages approximate the same original flow but with only the second stage probing the constraints via randomization, is more computationally efficient than the corresponding one-stage approach.*”
>
>    3. “The REINFORCE gradient underpinning FM-RE is known to exhibit high variance with sparse binary rewards, yet variance is neither measured nor discussed.”
>
> We thank the reviewer for pointing this out. We added Eq. (22) and (23) to validate the correctness of adding a baseline in our cases. Then we added the analyzes to mean gradient norm and gradient variance in D.1.2. We refer the reviewer to the revised manuscript for updates.
>
>    4. “The SWD confidence intervals in Table 5 overlap between FM-DD/FM-RE and vanilla FM in multiple settings, casting doubt on whether some distributional matching improvements are statistically meaningful.”
>
> We agree with the reviewer that FM-DD/FM-RE **cannot** improve distributional matching when compared with FM. However, we need to clarify that we did not claim and should not expect FM-DD/FM-RE to drastically improve distributional matching. The reason is that the proposed methods share the same FM objective with vanilla FM.
>
> What we actually claim is “the proposed approaches achieve significant gains in terms of constraint satisfaction while also matching the target distributions”(in the abstract). Table 5 (Table 6 in the updated manuscript) aims to show that the proposed methods and FM can have similar performance in distributional matching, while the proposed methods can achieve higher constraint satisfaction rates. For a concrete example, we can refer to Fig. 1, while all methods both generate similar distributions (all do well in distributional matching), FM-DD/FM-RE violates the constraints less.

---

> ### Author Response · Authors · 2026-03-13
>
> Responses to the requested changes:
>
>    1. We refer the reviewer to our response to comment 1.
>
>    2. The applications mentioned in the introduction (watermark generation, fluid dynamics, and high-quality image generation) are drawn from representative **sample-wise constraints** discussed in [1,2,3]. In those works, the constraint sets possess additional structure that can be exploited—for example, convex constraint sets in [1], differentiable distance functions to the constraint set in [2], or connected sets with known boundary information in [3].
>
> While FM-RE is, in principle, applicable to these tasks, including such experiments would not necessarily  highlight further what the main novelty of our method is. In particular, FM-RE does not rely on the convexity of the constraints set, availability of a differentiable distance, and other additional structures assumed for these settings. Instead, the goal of our experimental section is to evaluate the method in more general settings where these favorable assumptions are absent, thereby demonstrating its broader applicability.
>
> In the revision,  we change a general application from “high-quality image generation [3, 4]” to **“image generation with certain attributes”** to highlight our experimental settings.
>
>    3. The reviewer’s statement “This is tantamount to learned noise injection rather than constrained generation.” is not entirely accurate. Mathematically, the FM term $\mathbb{E}[\\| u_\theta(X_t, t) - \frac{d}{dt} \Psi_t(X_0, X_1)\\|^2]$ still exists. Therefore, the objective still requires the model’s estimated velocity  to be close to the original flow, i.e., learning to generate reasonable images. Moreover, merely learning to inject noise with the objective to mislead the classifier will not keep the images visually reasonable. The reason is that one can add arbitrarily large noise (without considering the original flow) to heavily mislead the classifier.
>
> In this case, we define an image to satisfy the constraint if it is an adversarial example, i.e., if the classifier assigns a label that is different from the ground truth. Therefore, the accuracy of the classifier is the constraint violation rate. To make this clear, we make the following change in Fig. 4’s caption (please see the revised version),
>
>  “*The accuracy of LeNet-$5$ on MNIST (i.e., the constraint violation rate $\mathbb{P}(X_1\notin \mathcal{C})$) drops from $99.1\\%$ to $18.7\\%$. The accuracy of ResNet-$50$ on CIFAR-$10$ (i.e., the constraint violation rate $\mathbb{P}(X_1\notin \mathcal{C})$) drops from $95.3\\%$ to $28.2\\%$.*”
>
>    4. Compared with FM-RE, FM-DD’s advantages lie on: (1) FM-DD does not need to introduce randomness in the sampling trajectory. (2) FM-DD’s gradient update is more stable since it optimizes a smooth, differentiable distance penalty. (3) FM-DD empirically achieves better constraint satisfaction performance than FM-RE, as shown in Table 2.
>
> While the limitation of FM-DD compared with FM-RE is that FM-DD can only apply to the cases, in which a differentiable distance to the constraint set is available. For example, FM-DD cannot apply to the tasks in Sec, 6.2 and 6.3.
>
> In conclusion, we can choose between FM-DD and FM-RE based on the following principle: If a differentiable distance to the constraint set is available and easy to estimate, FM-DD is preferred. If the membership oracle to the constraint set is available and the differentiable distance is not available, FM-DD is not applicable and FM-RE is preferred.
>
> We made the following changes in Sec. 7,
>
> “*In this paper, we present general constraint-aware FM frameworks: FM-DD and FM-RE. FM-DD incorporates a smooth, differentiable distance-based penalty for constraint violations, while FM-RE enforces constraint satisfaction by randomization directed to explore the constraints via access to a membership oracle. Compared to FM-RE, FM-DD yields empirically higher constraint satisfaction but requires access to a differentiable distance function to the constraint set. Thus, FM-DD is preferable when such distances are available, whereas FM-RE is suited to general settings with a membership oracle for the constraint set.*”
>
>    5. We refer the reviewer to our response to comment 3.
>
>    6.  We agree with the review’s suggestions to move Table 4 to the main text. We made a new subsection at the end of Sec. 6 to discuss the limitations of the proposed methods, including the computational efficiencies. We also add the computational time of other related approaches (RFM, MDM and PDM) for a more comprehensive comparison. We refer the reviewer to Sec. 6.5 and Table 4 in the revised manuscript for updates.
>
>
>
> [1] Guan-Horng Liu et al. Mirror diffusion models for constrained and watermarked generation.
>
> [2] Berthy T. Feng et al. Neural approximate mirror maps for constrained diffusion models
>
> [3] Tianyu Xie et al. Reflected flow matching
>
> [4] Aaron Lou and Stefano Ermon. Reflected diffusion models

---

### Decision · Action_Editor_hefS · 2026-04-07

**Recommendation:** Accept as is

**Audience:**

Yes

**Audience Explanation:**

Constrained sampling is an important problem which this paper addresses with thought and care.

**Claims And Evidence:**

Yes

**Claims Explanation:**

While the reviewers agree on the limited experimental scope, there is a consensus that the evidence supports the algorithmic claims made by the authors.